# Inconsistency of ammonium-sulfate aerosol ratios with thermodynamic models in the eastern US: a possible role of organic aerosol

Rachel F. Silvern[1], Daniel J. Jacob[1,2], Patrick S. Kim[1], Eloise A. Marais[2,a], Jay R. Turner[3], Pedro Campuzano-Jost[4,5], and Jose L. Jimenez[4,5]

[1]Department of Earth and Planetary Sciences, Harvard University, Cambridge, MA, USA
[2]John A. Paulson School of Engineering and Applied Sciences, Harvard University, Cambridge, MA, USA
[3]Department of Energy, Environmental and Chemical Engineering, Washington University, St. Louis, Missouri, USA
[4]Cooperative Institute for Research in Environmental Sciences, University of Colorado Boulder, Boulder, CO, USA
[5]Department of Chemistry and Biochemistry, University of Colorado Boulder, Boulder, CO, USA
[a]now at: School of Geography, Earth and Environmental Sciences, University of Birmingham, Edgbaston, UK

*Correspondence to:* Rachel F. Silvern (rsilvern@g.harvard.edu)

**Abstract.** Thermodynamic models predict that sulfate aerosol (S(VI) $\equiv$ $H_2SO_4$(aq) + $HSO_4^-$ + $SO_4^{2-}$) should take up available ammonia ($NH_3$) quantitatively as ammonium ($NH_4^+$) until the ammonium sulfate stoichiometry ($NH_4$)$_2SO_4$ is close to being reached. This uptake of ammonia has important implications for aerosol mass, hygroscopicity, and acidity. When ammonia is in excess, the ammonium-sulfate aerosol ratio $R = [NH_4^+]/[S(VI)]$ should approach 2 with excess ammonia remaining in the gas phase. When ammonia is in deficit, it should be fully taken up by the aerosol as ammonium and no significant ammonia should remain in the gas phase. Here we report that sulfate aerosol in the eastern US in summer has a low ammonium-sulfate ratio despite excess ammonia, and we show that this is at odds with thermodynamic models. The ammonium-sulfate ratio averages only 1.04 ± 0.21 mol mol$^{-1}$ in the Southeast, even though ammonia is in large excess as shown by the ammonium-sulfate ratio in wet deposition and by the presence of gas-phase ammonia. It further appears that the ammonium-sulfate aerosol ratio is insensitive to the supply of ammonia, remaining low even as the wet deposition ratio exceeds 6 mol mol$^{-1}$. While the ammonium-sulfate ratio in wet deposition has increased by 5.8% a$^{-1}$ from 2003 to 2013 in the Southeast US, consistent with $SO_2$ emission controls, the ammonium-sulfate aerosol ratio has decreased by 1.4-3.0% a$^{-1}$. Thus the aerosol is becoming more acidic even as $SO_2$ emissions decrease and while ammonia emissions are staying constant; this is incompatible with simple sulfate-ammonium thermodynamics. A tentative explanation is that sulfate particles are increasingly coated by organic material, retarding the uptake of ammonia. Indeed, the ratio of organic aerosol (OA) to sulfate in the Southeast increased from 1.1 to 2.4 g g$^{-1}$ over the 2003-2013 period as sulfate decreased. We implement a simple kinetic mass transfer limitation for ammonia uptake to sulfate aerosols in the GEOS-Chem chemical transport model and find that we can reproduce both the observed ammonium-sulfate aerosol ratios and the concurrent presence of gas-phase ammonia. If sulfate aerosol becomes more acidic as OA/sulfate ratios increase, then controlling

SO$_2$ emissions to decrease sulfate aerosol will not have the co-benefit of suppressing acid-catalyzed secondary organic aerosol (SOA) formation.

## 1. Introduction

Sulfuric acid (H$_2$SO$_4$) produced in the atmosphere by oxidation of sulfur dioxide (SO$_2$) has very low vapor pressure in the presence of water vapor and immediately forms aqueous sulfate aerosol, S(VI) $\equiv$ H$_2$SO$_4$(aq) + HSO$_4^-$ + SO$_4^{2-}$. This sulfate aerosol is a major component of fine particulate matter (PM$_{2.5}$, less than 2.5 μm diameter). The acid dissociation of sulfate is mostly driven by ammonia (NH$_3$) emitted from agriculture and natural sources and partitioning between the gas and aerosol phases (NH$_x$ $\equiv$ NH$_3$(g) + NH$_3$(aq) + NH$_4^+$). Depending on the supply of ammonia, sulfate aerosol may be speciated as sulfuric acid (H$_2$SO$_4$(aq)), ammonium bisulfate (NH$_4^+$, HSO$_4^-$), ammonium sulfate (2NH$_4^+$, SO$_4^{2-}$), and combinations in between. This speciation has important implications for aerosol mass, hygroscopicity, and acidity (Martin, 2000). When ammonia is in excess, standard thermodynamic models predict that sulfate aerosol should be mainly present as ammonium sulfate with an ammonium-sulfate ratio $R = $ [NH$_4^+$]/[S(VI)] approaching 2 on a molar basis (Seinfeld and Pandis, 2006). This thermodynamic behavior is indeed observed in a wide range of environments (Zhang et al., 2002; Martin et al., 2004; Yu et al., 2005). However, surface and aircraft observations in the Southeast US in summer find $R$ to be in the range 1.0-1.6 mol mol$^{-1}$ even with excess ammonia in the gas phase (Attwood et al., 2014; Guo et al., 2015; Kim et al., 2015). Here we examine the prevalence of this departure from expected thermodynamic behavior by analyzing aerosol and wet deposition data across the eastern US with focus on the Southeast, and we suggest a tentative explanation.

SO$_2$ emissions in the Southeast US declined by 63% from 2003 to 2013 due to regulatory controls on coal combustion (Hidy et al., 2014; US EPA, 2015). One would expect from standard sulfate-ammonium thermodynamics that this would result in an increase in the ammonium-sulfate ratio $R$. However, observations show that the sulfate and ammonium components of the aerosol decreased at similar rates over the period so that $R$ did not increase (Hand et al., 2012; Blanchard et al., 2013; Kim et al., 2015; Saylor et al., 2015; Weber et al., 2016), adding to the thermodynamic puzzle.

Weber et al. (2016) presented a detailed thermodynamic analysis of 1998-2013 observations of sulfate and ammonium aerosol and gas-phase ammonia at a rural site in the Southeast (Centreville, Alabama). They find a decrease in $R$ from 1.8 to 1.5 mol mol$^{-1}$ over the period even as the sulfate concentrations decrease, with significant ammonia (0.1-1 μg m$^{-3}$, ~ 0.1-1 ppb) remaining in the gas phase throughout the period. They show with the commonly used ISORROPIA II thermodynamic model (Fountoukis and Nenes, 2007) that the presence of this gas-phase ammonia is compatible with high aerosol acidity (pH 0-1.5) due to the semi-volatility of ammonia. However, their model calculations predict values

for $R$ in excess of 1.9 mol mol$^{-1}$, significantly higher than observed. As pointed out below, sulfate aerosol with $R$ below 1.8 mol mol$^{-1}$ should have very low ammonia vapor pressure ($<<$ 0.1 µg m$^{-3}$) according to ISORROPIA. There thus remains a difficulty in reconciling their simultaneous observations of significant gas-phase ammonia (indicating ammonia in excess) and low values of $R$ (indicating ammonia in deficit). A low value of $R$ could be explained if alkaline cations other than ammonium contributed to sulfate neutralization, or if part of S(VI) was in the form of organosulfates; however, observations in the Southeast US show that neither of these effects is significant (Budisulistiorini et al., 2015; Hettiyadura et al., 2015; Kim et al., 2015; Liao et al., 2015; Rattanavaraha et al., 2016). The chemical composition of individual sulfate particles may deviate from the bulk, but it is not clear how such inhomogeneity could explain the observed departure from simple thermodynamics.

Liggio et al. (2011) found in laboratory experiments that uptake of ammonia by sulfuric acid aerosol is hindered by the presence of organic gases, and proposed that competition for uptake between ammonia and organic gases slows down considerably the approach to thermodynamic equilibrium. Kim et al. (2015) hypothesized that this could explain the observations of low ammonium-sulfate ratios. Organic aerosol (OA) often dominates over sulfate (Zhang et al., 2007), and in particular in the Southeast US in summer where there is a large OA source from biogenic hydrocarbons (Kim et al., 2015; Marais et al., 2016a). Mixing of organic and sulfate aerosol may slow down mass transfer due to phase separation, in which the organic aerosol fraction coats the predominantly aqueous inorganic core, as has been observed in many laboratory studies of organic-ammonium-sulfate particles (Anttila et al., 2007; Ciobanu et al., 2009; Bertram et al., 2011; Koop et al., 2011; You et al., 2013) as well as in the field in the Southeast US (You et al., 2012).

## 2. Thermodynamics of the H$_2$SO$_4$-NH$_3$ system

H$_2$SO$_4$-HNO$_3$-NH$_3$ mixtures in the atmosphere form sulfate-nitrate-ammonium (SNA) aerosol following well-established thermodynamic rules (Martin, 2000). Nitrate partitions into the aerosol only when ammonia is in excess of sulfate and temperatures are low (Ansari and Pandis, 1998; Park et al., 2004). Nitrate is a negligibly small component of the aerosol in the Southeast US in summer (Ford and Heald, 2013; Kim et al., 2015). Here we focus on the H$_2$SO$_4$-NH$_3$ system, ignoring HNO$_3$ which is unimportant for our argument.

The thermodynamics of the H$_2$SO$_4$-NH$_3$ system is determined by the supply of total sulfate (S(VI)) and ammonia (NH$_x$), relative humidity (RH), and temperature ($T$). Here we consider an aqueous aerosol (which may be metastable) in equilibrium with the gas phase. S(VI) is exclusively in the aerosol phase as the sum of H$_2$SO$_4$(aq) and its acid dissociation products. NH$_x$ partitions between the gas and the aerosol

phase as $NH_x \equiv NH_3(g) + NH_3(aq) + NH_4^+$. $NH_3(aq)$ is a negligibly small component of $NH_x$ under all atmospheric conditions.

Figure 1 (left panel) shows the ammonium-sulfate ratio $R = [NH_4^+]/[S(VI)]$ and the aerosol pH at thermodynamic equilibrium in the $H_2SO_4$-$NH_3$ system, calculated by ISORROPIA II as a function of the input ratio $[NH_x]/[S(VI)]$. The calculations are for an aqueous aerosol with RH = 70% and $T$ = 298 K, typical of conditions in the Southeast US in summer. Curves are shown for $[S(VI)]$ = 1 and 5 $\mu g\ m^{-3}$, representing a range of moderately polluted conditions. The ammonium-sulfate ratio $R$ closely follows the total $[NH_x]/[S(VI)]$ molar ratio up to a value of 1.8 (depending on the S(VI) concentration), and from there asymptotically approaches 2 as ammonia becomes in excess of sulfuric acid. Gas-phase ammonia is less than 0.01 $\mu g\ m^{-3}$ for $[NH_x]/[S(VI)]$ below 2, at odds with the Weber et al. (2016) observations of $R < 1.8$ mol mol$^{-1}$ with $[NH_3(g)] > 0.1\ \mu g\ m^{-3}$. The aerosol pH calculated by ISORROPIA remains low (0.5-1.75) even with ammonia in large excess. This was previously pointed out by Guo et al. (2015) and Xu et al. (2015), and reflects the small aerosol liquid water content combined with the limited solubility of ammonia. It explains why gaseous ammonia is observed in the Southeast US at levels consistent with thermodynamic models even when the aerosol is acidic according to the pH metric (Nowak et al., 2006; Guo et al., 2015; Weber et al., 2016).

The right panel of Figure 1 shows the same thermodynamic analysis using the Extended Aerosol Inorganic Model (E-AIM; Wexler and Clegg, 2002), which makes fewer assumptions than ISORROPIA II. We use E-AIM IV (Friese and Ebel, 2010), available interactively from http://www.aim.env.uea.ac.uk/aim/aim.php. E-AIM and ISORROPIA predict similar pH values, as pointed out by Hennigan et al. (2015), but E-AIM is much slower than ISORROPIA in approaching the $R = 2$ asymptote. Thus the Weber et al. (2016) observations could be accommodated by the E-AIM thermodynamic model in the $[NH_x]/[S(VI)] > 2$ regime. However, E-AIM still cannot reproduce the much lower values of $R$ observed at other sites in the Southeast nor can it explain the trend of decreasing $R$ as $SO_2$ emissions decrease. It has been shown that ammonium-sulfate aerosol ratios are not a simple proxy for aerosol pH (Hennigan et al., 2015). Here, we focus only on the the measureable quantity, $R$, and we describe these observations further in what follows.

## 3. Ammonium-sulfate ratios in aerosol and precipitation

Figure 2 (top left panel) shows the $NH_3/SO_2$ molar emission ratio for the eastern US in summer 2013. Here and throughout this paper, mean ratios are presented as the ratios of the mean quantities. The emissions are from the 2011 National Emission Inventory (NEI) of the US Environmental Protection Agency (EPA), scaled to 2013 as described by Kim et al. (2015). There is good confidence in US ammonia emissions, which agree within 20% in independent bottom-up and top-down estimates (Paulot et al., 2014).

Most of the domain has an emission ratio higher than 2, indicating excess ammonia. Total emission in the eastern US (domain of Figure 2, east of 95°W) is 45 Gmol $NH_3$ and 15 Gmol $SO_2$ for the three summer months, corresponding to a $NH_3/SO_2$ emission ratio of 3.0 mol mol$^{-1}$. About a third of emitted $SO_2$ may be removed by dry deposition rather than produce sulfate (Chin and Jacob, 1996), so that ammonia would be even more in excess, although 20-30% of ammonia may also be removed by dry deposition in the eastern US (Li et al., 2016).

The excess of ammonia is apparent in the $[NH_4^+]/[S(VI)]$ wet deposition flux data from the National Atmospheric Deposition Program (NADP) National Trends Network (NTN; http://nadp.sws.uiuc.edu/data/ntn/), shown in the top right panel of Figure 2. Both aerosol $NH_4^+$ and $NH_3(g)$ are efficiently scavenged by precipitation, so that the ammonium wet deposition flux relates to total ammonia emission. Similarly, both sulfate and $SO_2$ are efficiently scavenged so that the sulfate wet deposition flux relates to total $SO_2$ emission. The mean ammonium-sulfate ratio in the wet deposition flux data over the eastern US domain of Figure 2 is 3.0 mol mol$^{-1}$, again indicating ammonia in excess. Values less than 2 are mainly confined to the industrial Midwest (where the $NH_3/SO_2$ emission ratio is low) and to the Gulf Coast where precipitation may have a strong maritime influence. This excess of ammonia in the emission and wet deposition data is consistent with general observations of significant gas-phase ammonia concentrations at Southeast US sites (You et al., 2014; Guo et al., 2015; Saylor et al., 2015; Weber et al., 2016).

The bottom panels of Figure 2 show the ammonium-sulfate ratio in aerosol data from EPA's Chemical Speciation Network (CSN; Solomon et al., 2014), the Southeastern Aerosol Research and Characterization Study (SEARCH; Edgerton et al., 2005), and the Southern Oxidant and Aerosol Study (SOAS; Hu et al., 2015). The bottom right panel shows an alternate estimate of the ratio as $R_N = ([NH_4^+]-[NO_3^-])/[S(VI)]$ in order to remove the component of ammonium associated with ammonium nitrate (Weber et al, 2016). We expect $R_N$ and $R$ to bracket the effective ammonium-sulfate ratio, depending on whether aerosol nitrate is associated with ammonium or with other cations. The difference between the two is small in the Southeast US where the contribution of nitrate in summer is very small (Ford and Heald, 2013; Kim et al., 2015). Nitrate at the ensemble of Southeast US sites averages $0.25 \pm 0.08$ µg m$^{-3}$ in summer 2013, representing less than 4% of $PM_{2.5}$ mass. Aerosol amines are present in low concentrations in the Southeast US (You et al., 2014) and concentrations of alkaline cations other than ammonium (e.g., $Ca^{2+}$, $Mg^{2+}$) are also too low to affect significantly the charge balance, as previously shown by Kim et al. (2015). Concentrations of these other alkaline cations are reported at the CSN sites, and we find for the ensemble of CSN sites in Figure 2 that they would modify $R$ on average by 0.11 mol mol$^{-1}$.

Results in Figure 2 show that the ammonium-sulfate aerosol ratio is consistently well below 2, which is thermodynamically inconsistent with the presence of excess ammonia. The mean ($\pm$ standard

deviation) aerosol ratios for CSN sites in the domain of Figure 2 are $R_N = 1.08 \pm 0.26$ mol mol$^{-1}$ and $R = 1.44 \pm 0.34$ mol mol$^{-1}$. Mean values for the five SEARCH sites in the Southeast are $R_N = 1.52 \pm 0.18$ mol mol$^{-1}$ and $R = 1.62 \pm 0.17$ mol mol$^{-1}$. Aerosol mass spectrometer (AMS) measurements for the SOAS ground site in Centreville, Alabama in June-July 2013 give $R_N = 0.85 \pm 0.31$ mol mol$^{-1}$ and $R = 0.93 \pm 0.29$ mol mol$^{-1}$, consistent with Particle into Liquid Sampler (PILS) measurements at the same site (Guo et al., 2015). AMS measurements onboard the NASA SEAC[4]RS aircraft (Wagner et al., 2015) in the Southeast US boundary layer (below 2 km altitude) in August 2013 averaged $R_N = 1.29 \pm 0.44$ mol mol$^{-1}$ and $R = 1.39 \pm 0.52$ mol mol$^{-1}$. Low values of $R$ are consistent with the lack of nitrate in the aerosol (Guo et al., 2015; Weber et al., 2016).

One sigma ($1\sigma$) precision estimates for CSN network sulfate and ammonium aerosol concentrations are 6% and 8% respectively (Flanagan et al., 2006). For the SEARCH network the precision statistics are reported as median absolute differences (Edgerton et al., 2005). Assuming the measurement error is normally distributed these precision statistics can be converted to $1\sigma$ values (Rousseeuw and Croux, 1993) of 3% for sulfate and 5% for ammonium, respectively. The corresponding propagated uncertainties for $R$ are 0.1 mol mol$^{-1}$ (CSN) and 0.06 mol mol$^{-1}$ (SEARCH).

Differences in ammonium filter measurement methods between the CSN and SEARCH networks likely account for the higher values of $R$ at the SEARCH sites. CSN samples for ion analysis are collected using a nylon filter downstream of a magnesium oxide denuder (Solomon et al., 2014). The use of a single nylon filter is prone to a negative bias because of volatilization losses of ammonia from ammonium nitrate (Yu et al., 2006). SEARCH samples for ion analysis are collected using a Teflon/nylon filter pack downstream of sodium bicarbonate and citric acid denuders. Best-estimate ammonium concentrations are calculated using the nonvolatile ammonium from the Teflon filter plus the stoichiometric ammonium associated with the nitrate measured on the nylon backup filter; this approach assumes that the particles volatilizing from the Teflon front filter are solely ammonium nitrate (Edgerton et al., 2005). Comparing these methods, CSN could be prone to a positive artifact because an acid-coated denuder is not used to remove gaseous ammonia but this bias is likely outweighed by the negative artifact when ammonium nitrate volatilizes and the resulting ammonia is not quantitatively retained by the nylon filter. However, Yu et al. (2006) showed in summertime observations at Great Smoky Mountains National Park (Tennessee) that ammonium losses could not be explained by particulate nitrate and suggested that organic ammonium salts could contribute to measured ammonium. If organic ammonium salts were retained on the filters at CSN or SEARCH sites, this would mean a lower effective ammonium-sulfate ratio.

Figure 2 shows more acidic conditions (lower ammonium-sulfate ratios) in the Southeast than in the Northeast. The Southeast CSN sites (south of 37°N) have $R_N = 0.81 \pm 0.21$ mol mol$^{-1}$ and $R = 1.04 \pm 0.21$

mol mol$^{-1}$, while the Northeast sites have $R_N = 1.17 \pm 0.22$ mol mol$^{-1}$ and $R = 1.57 \pm 0.27$ mol mol$^{-1}$. The difference between $R_N$ and $R$ is less in the Southeast because the contribution of nitrate to aerosol composition is very small. The same regional mean pattern is seen in the ammonium-sulfate wet deposition flux ratios ($2.23 \pm 0.80$ mol mol$^{-1}$ in Southeast, $2.99 \pm 1.33$ mol mol$^{-1}$ in Northeast). The emission ratio NH$_3$/SO$_2$ is 3.28 mol mol$^{-1}$ in the Southeast and 2.69 mol mol$^{-1}$ in the Northeast, but SO$_2$ may be oxidized to sulfate more efficiently in the Southeast because of higher oxidant concentrations and longer residence times.

Figure 3 shows the relationship in the Southeast between aerosol and wet deposition ammonium-sulfate ratios for collocated sites, compared to thermodynamic predictions from E-AIM IV and ISORROPIA II. Here we take the observed wet deposition ammonium-sulfate ratio to be a measure of the [NH$_x$]/[S(VI)] ratio input to thermodynamic models, which should be qualitatively correct. We see that the observed ammonium-sulfate aerosol ratio does not follow thermodynamic predictions and shows no correlation with the wet deposition ammonium-sulfate ratio. The aerosol ratio remains between 0.92 ($R_N$) and 1.15 mol mol$^{-1}$ ($R$) even as the wet deposition ratio exceeds 6 mol mol$^{-1}$.

The departure of the ammonium-sulfate aerosol ratio from thermodynamic predictions is also apparent in observed long-term trends. Figure 4 shows 2003-2013 trends in the Southeast US in summer at CSN and NADP sites. Sulfate wet deposition fluxes and aerosol concentrations both decrease by 6-8% a$^{-1}$, consistent with the trend in SO$_2$ emissions (Hand et al., 2012). There is no significant change in NH$_4^+$ wet deposition fluxes, as expected from constant NH$_3$ emissions during this period (Xing et al., 2013; Saylor et al., 2015). However, aerosol ammonium decreases at a rate similar to sulfate (-8.5% a$^{-1}$). Figure 5 shows trends at SEARCH sites, which also show aerosol sulfate and ammonium declining at a similar rate (-9.2% a$^{-1}$ and -9.1% a$^{-1}$ respectively), consistent with results previously shown by Weber et al. (2016). Such a parallel decrease of sulfate and ammonium would be expected only if the ammonium-sulfate aerosol ratio was very close to the asymptotic value of 2, in which case aerosol ammonium would be limited by the supply of sulfate; however, the observed ammonium-sulfate aerosol ratios are much lower. Thermodynamic predictions in Figures 1 and 3 show that as the supply of sulfate decreases relative to NH$_x$, the ammonium-sulfate aerosol ratio should increase. Marais et al. (2016b) shows that standard thermodynamics predict a significant decrease in aerosol acidity in response to the decrease in sulfate. However, the opposite is observed. The ammonium-sulfate aerosol ratio decreases by 3.0% a$^{-1}$ at CSN sites and 1.4% a$^{-1}$ at SEARCH sites, consistent with Weber et al. (2016) who showed a decline in the ratio by 1.4% a$^{-1}$ for 1998-2013 aerosol observations at the Centreville, AL SEARCH site. Thus the aerosol is becoming more acidic even as SO$_2$ emission decreases.

## 4. Possible mass transfer limitation by organic aerosol?

One possible explanation for the low and decreasing ammonium-sulfate aerosol ratios observed in the Southeast US is that organic aerosol (OA) may affect SNA thermodynamics or slow down the achievement of SNA thermodynamic equilibrium. We propose a tentative explanation of the observations based on the latter. As shown in Figure 5, the OA/S(VI) ratio in the Southeast increases rapidly over the 2003-2013 period in response to decreasing $SO_2$ emissions. Liggio et al. (2011) found in laboratory experiments using ambient air that uptake of ammonia by acidic sulfate aerosol is slowed by the uptake of organic gases. Measured timescales to reach equilibrium for experiments where organics were present were on the order of hours, significantly longer than the timescale of seconds measured for organic-free experiments. Daumer et al. (1992) previously noted a retardation in ammonia uptake for sulfuric acid particles coated with organic films. Liggio et al. (2011) reported reactive uptake coefficients ($\gamma$) for ammonia as a function of the mass ratio of OA to sulfate in their experiments. $\gamma$ is defined as the probability that an ammonia molecule impacting the acidic sulfate aerosol will be taken up as $NH_4^+$. For OA to sulfate mass ratios of 0.14, 0.25, and 0.55, Liggio et al. (2011) reported $\gamma$ values of $4\times10^{-3}$, $2\times10^{-4}$, and $5\times10^{-4}$ respectively, in contrast to $\gamma \approx 1$ for organic-free experiments.

The results of Liggio et al. (2011) suggest a possible mass transfer limitation to ammonia uptake by the aerosol phase dependent on the local OA concentration. This might be explained by an OA surfactant effect or other phase separation. Laboratory studies have shown liquid-liquid phase separation of organic-ammonium-sulfate particles for oxygen to carbon elemental ratios (O:C) $\leq$ 0.8 (You and Bertram, 2015). Boundary layer observations from the SEAC$^4$RS aircraft campaign over the Southeast US in summer 2013 indicate a mean O:C ratio of $0.75 \pm 0.22$, suggesting that phase separation may occur.

The values of $\gamma$ reported by Liggio et al. (2011) can be used to describe a kinetic limitation to ammonia uptake where the net uptake of $NH_3(g)$ by the SNA aerosol is given by

$$-\frac{d[NH_3(g)]}{dt} = k\left([NH_3(g)] - [NH_3(g)]_{eq}\right) \tag{1}$$

The mass transfer rate constant $k$ [s$^{-1}$] is applied in equation (1) to the difference between the local concentration of $NH_3(g)$ and that computed from SNA thermodynamic equilibrium. $k$ is related to $\gamma$ (Jacob, 2000) by:

$$k = \int_0^\infty 4\pi a^2 \left(\frac{a}{D_g} + \frac{4}{\gamma v}\right)^{-1} n(a)\, da \tag{2}$$

where $a$ is the wet aerosol radius, $D_g$ is the gas phase diffusion coefficient, $v$ is the mean molecular speed, and $n(a)$ is the number size distribution of sulfate aerosol.

We implemented this crude kinetic limitation to ammonia uptake by SNA aerosol into the GEOS-Chem chemical transport model (CTM) version 9-02, previously applied by Kim et al. (2015) to simulation of aerosol observations from the NASA SEAC[4]RS aircraft campaign over the Southeast US in summer-fall 2013 (Toon et al., 2016). The simulation includes detailed oxidant-aerosol chemistry as described by Kim et al. (2015) and Travis et al. (2016). Ammonia and $SO_2$ emissions are from the EPA National Emission Inventory for 2011 modified for 2013, with the emission ratios of Figure 2. SNA aerosol thermodynamics follows ISORROPIA II. ISORROPIA II in GEOS-Chem uses the metastable phase state in which the aerosol phase is always aqueous. The standard GEOS-Chem model assumes that SNA aerosol is in thermodynamic equilibrium at all times. Here we introduce the kinetic limitation to ammonia uptake described above.

Kim et al. (2015) presented detailed comparisons of results from the standard GEOS-Chem model assuming SNA thermodynamic equilibrium to aerosol observations collected from aircraft, surface sites, and satellites during SEAC[4]RS. They showed that GEOS-Chem simulates successfully and without bias the observed sulfate and OA concentrations from the CSN network and the SEAC[4]RS aircraft. However, their simulated ammonium concentrations were too high. Figure 6 shows that the ammonium-sulfate aerosol ratio in the standard model over most of the eastern US is close to 2 mol mol$^{-1}$, as expected from SNA thermodynamics with ammonia in excess; but the observed ratios are much lower. A reduced major axis (RMA) regression for the SEAC[4]RS flight tracks gives a standard model ratio of $2.08 \pm 0.02$ mol mol$^{-1}$, whereas the observations give a ratio of $1.21 \pm 0.08$ mol mol$^{-1}$. The standard model ratio is slightly in excess of 2 because of the contribution of nitrate aerosol.

Figure 7 compares the gas-phase ammonia concentrations in the standard model to observations at the SEARCH sites. The model simulates concentrations of 0.05-1.2 µg m$^{-3}$, biased low by 44%. The standard model reproduces the mean observed wet deposition fluxes of ammonium over the Southeast US in summer ($0.15 \pm 0.10$ kg N ha$^{-1}$ month$^{-1}$ modeled, $0.19 \pm 0.12$ kg N ha$^{-1}$ month$^{-1}$ observed) showing that uncertainty in ammonia emissions is not sufficient to explain the underestimate in gas-phase ammonia. The presence of gas-phase ammonia in the standard model is contingent on excess ammonia and an ammonium-sulfate aerosol ratio close to 2 (Figure 1). The problem is thus to explain the joint presence of gas-phase ammonia and low ammonium-sulfate aerosol ratios in the observations. Kinetic mass transfer limitation of ammonia uptake by SNA aerosols following equations (1) and (2) can solve that problem, as shown in Figures 6 and 7. Observed OA to sulfate mass ratios in the eastern US in summer 2013 average $1.89 \pm 0.83$ g g$^{-1}$ at CSN sites and $2.44 \pm 1.11$ g g$^{-1}$ at SEARCH sites (Figure 5), exceeding the maximum ratio of 0.55 reported by Liggio et al. (2011). Similarly, OA to sulfate ratios in the model are much greater than 0.55

throughout the eastern US boundary layer. We assume in GEOS-Chem that $\gamma = 5 \times 10^{-4}$ wherever the OA to sulfate ratio exceeds 0.55, following Liggio et al. (2011). This implies a timescale of over one day for ammonia to reach equilibrium.

GEOS-Chem with this kinetic limitation captures the low ammonium-sulfate ratio in the CSN observations in the Southeast ($R = 1.14 \pm 0.21$ observed, $1.02 \pm 0.10$ modeled) and overcorrects in the Northeast ($R = 1.51 \pm 0.21$ observed, $1.06 \pm 0.21$ modeled). OA/S(VI) concentration ratios are lower in the Northeast and so the kinetic limitation could be less. The ammonium-sulfate ratio in the SEAC$^4$RS aircraft observations is also better simulated as indicated by RMA regressions for the flight tracks in Figure 6 (1.39 $\pm$ 0.03 modeled, $1.21 \pm 0.08$ observed). The model is further successful at reproducing the gas-phase ammonia concentrations at the SEARCH sites, with no significant bias. In the absence of kinetic limitation, such low ammonium-sulfate ratios would be incompatible with the presence of significant gas-phase ammonia concentrations (Figure 1).

## 5. Conclusions

Observation networks in the eastern US show low ammonium-sulfate aerosol ratios even when total ammonia is in large excess. This departs from expected $H_2SO_4$-$NH_3$ thermodynamic equilibrium and has important implications for aerosol mass, hygroscopicity, and acidity. The ammonium-sulfate ratio $R = [NH_4^+]/[S(VI)]$ averages 1.04 mol mol$^{-1}$ in the Southeast and 1.57 mol mol$^{-1}$ in the Northeast in summer, even though ammonia is in excess as indicated by the wet deposition flux ratios and by the observations of gas-phase ammonia. Observed long-term trends for 2003-2013 show that aerosol sulfate and ammonium decreased together in response to $SO_2$ emission controls, whereas one would thermodynamically expect the ammonium-sulfate ratio to increase. In fact, the ammonium-sulfate ratio decreased by 1-3% a$^{-1}$ during the 2003-2013 period while $SO_2$ emissions decreased.

There appears to be a fundamental problem in reconciling from a thermodynamic perspective the joint observations of gas-phase ammonia and low ammonium-sulfate ratios. We suggest that this apparent departure from thermodynamic behavior may be caused by an elevated and increasing organic aerosol (OA) mass fraction, modifying or retarding the achievement of $H_2SO_4$-$NH_3$ thermodynamic equilibrium. Laboratory experiments by Liggio et al. (2011) indicate that the reactive uptake coefficient ($\gamma$) for uptake of ammonia by sulfate aerosol decreases greatly in the presence of OA. Implementation of a crude representation of this kinetic limitation in the GEOS-Chem chemical transport model greatly improves the agreement of the model with surface and aircraft observations of the ammonium-sulfate ratio in the eastern US, and also simulates successfully the observed gas-phase ammonia concentrations. Better understanding of OA effects on sulfate aerosol thermodynamics is needed. In addition to the phase separation hypothesis

explored here, it has also been shown experimentally that reactions between ammonia and organics can occur (Liu et al., 2015) with similar uptake coefficients to those measured by Liggio et al. (2011). A mass transfer retardation of thermodynamic equilibrium may also have broader implications for the partitioning of semi-volatile species and for hygroscopicity. Previous work has shown good agreement between

5 observed and modeled nitrate partitioning during winter in the eastern US (Guo et al., 2016) and organics have not been shown to affect the uptake of water to the degree that it would be a limiting factor for particle growth (Wong et al., 2014). More work is needed to measure the sensitivity of semi-volatile species to the presence of organic aerosol versus other factors controlling partitioning such as temperature and relative humidity, and the implications for aerosol pH.

**Acknowledgments.** This work was funded by the Earth Science Division of the US National Aeronautics and Space Administration and by the US National Science Foundation. We thank Scot Martin (Harvard) for valuable discussions.

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

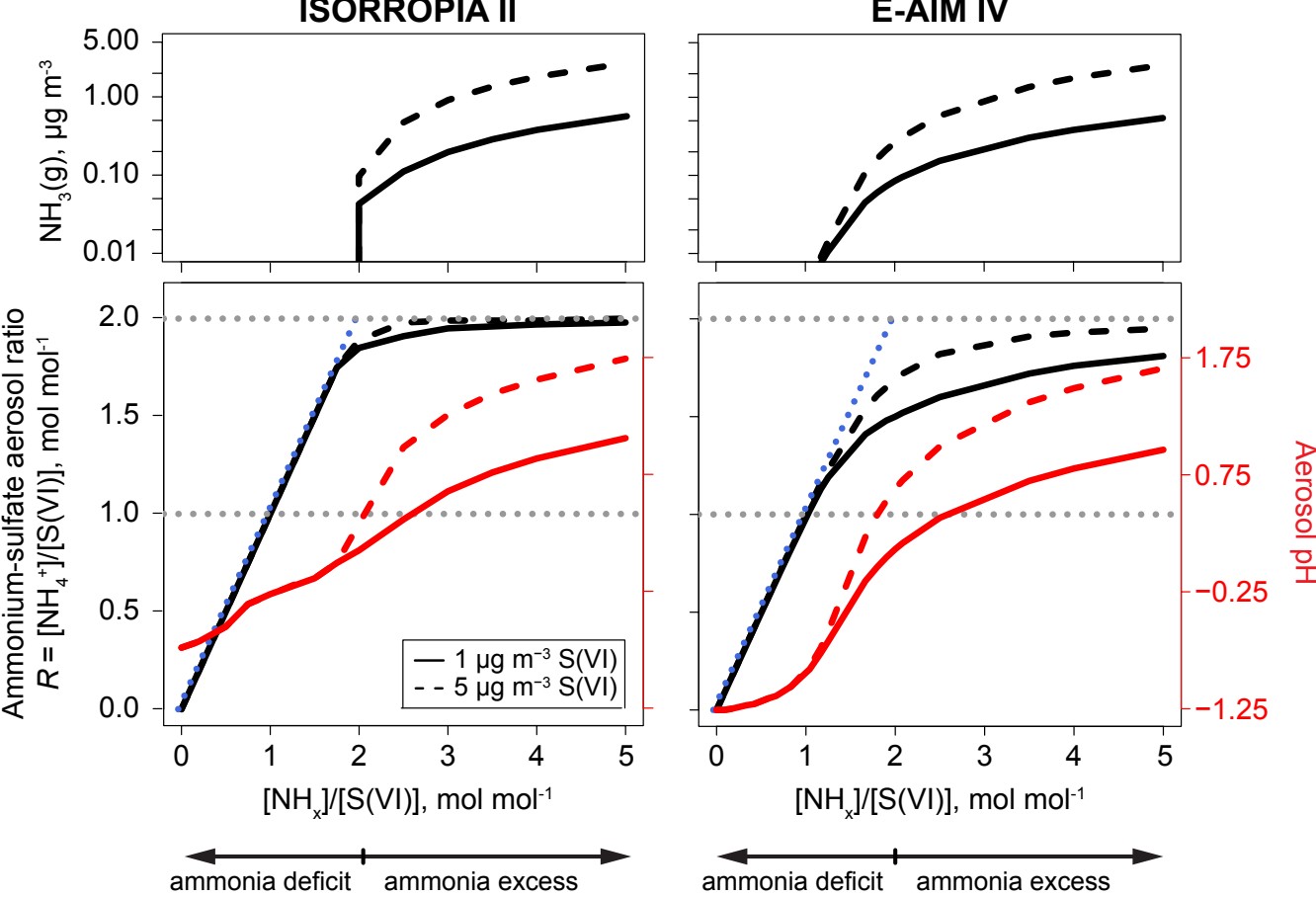

**Figure 1.** Thermodynamic properties of the sulfate-ammonium aerosol system as a function of the ratio of total ammonia ($NH_x \equiv NH_3(g) + NH_3(aq) + NH_4^+$) to total sulfate (S(VI)). A ratio lower than 2 indicates ammonia in deficit, a ratio higher than 2 indicates ammonia in excess. The figure plots the equilibrium gas-phase ammonia concentration (top panels), the ammonium-sulfate aerosol ratio ($R = [NH_4^+]/[S(VI)]$), and the aerosol pH. Values are computed with the thermodynamic models ISORROPIA II (left) and E-AIM IV (right) as a function of input [$NH_x$] with either 1 or 5 μg m$^{-3}$ S(VI). Both models are applied in the forward mode (total [S(VI)] and [$NH_x$] used as input) for a metastable aqueous aerosol with 70% relative humidity and 298 K. The 1:1 line for the relationship of $R$ to [$NH_x$]/[S(VI)] is shown in blue. The gray dotted lines show the ammonium-sulfate ratio $R = 1$ corresponding to $NH_4HSO_4$ and $R = 2$ corresponding to $(NH_4)_2SO_4$.

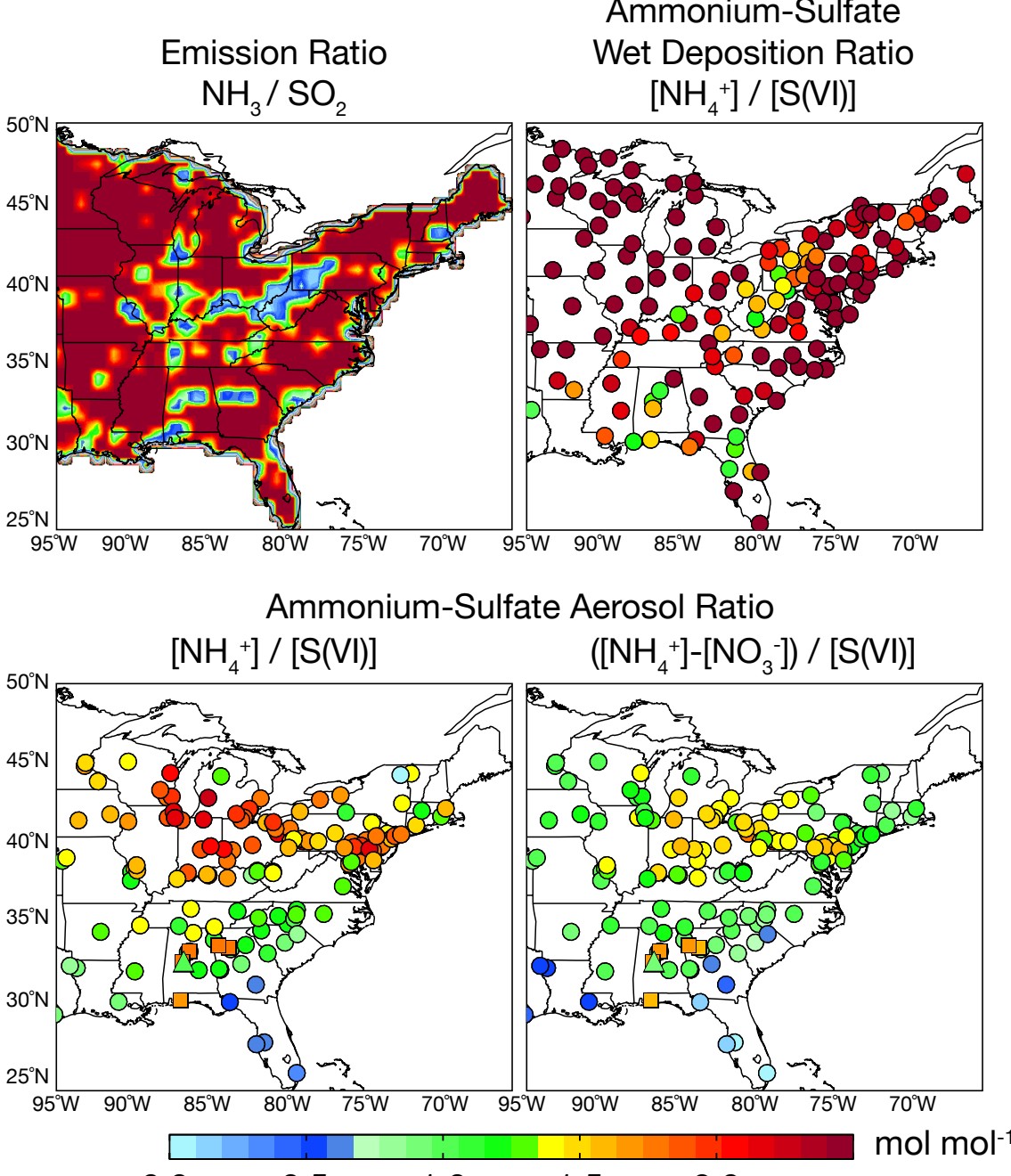

**Figure 2.** Ammonium-sulfate ratios in seasonally averaged data for the eastern US in summer 2013 (JJA). The top left panel shows the $NH_3/SO_2$ molar emission ratio from the EPA National Emission Inventory (NEI) on a $0.5°\times0.5°$ grid. The top right panel shows the $[NH_4^+]/[S(VI)]$ molar wet deposition flux ratio from the National Acid Deposition Network (NADP). The bottom panels show the molar aerosol ratios from the EPA Chemical Speciation Network (CSN; circles), the Southeastern Aerosol Research and Characterization Study (SEARCH; squares), and the Southern Oxidant and Aerosol Study (SOAS; triangles). Measurements from CSN and SEARCH are $PM_{2.5}$ and measurements from SOAS are $PM_1$. The bottom left panel shows $R = [NH_4^+]/[S(VI)]$ and the bottom right panel shows $R_N = ([NH_4^+]-[NO_3^-])/[S(VI)]$ where the subtraction of $[NO_3^-]$ is to remove the contribution of $NH_4^+$ to $NH_4NO_3$ aerosol. In both the wet deposition and aerosol data, we removed primary sea-salt sulfate on the basis of measured $Na^+$ as in Alexander et al. (2005); this represents a significant correction for coastal sites. Here and elsewhere, mean ratios are calculated as the ratios of the mean quantities.

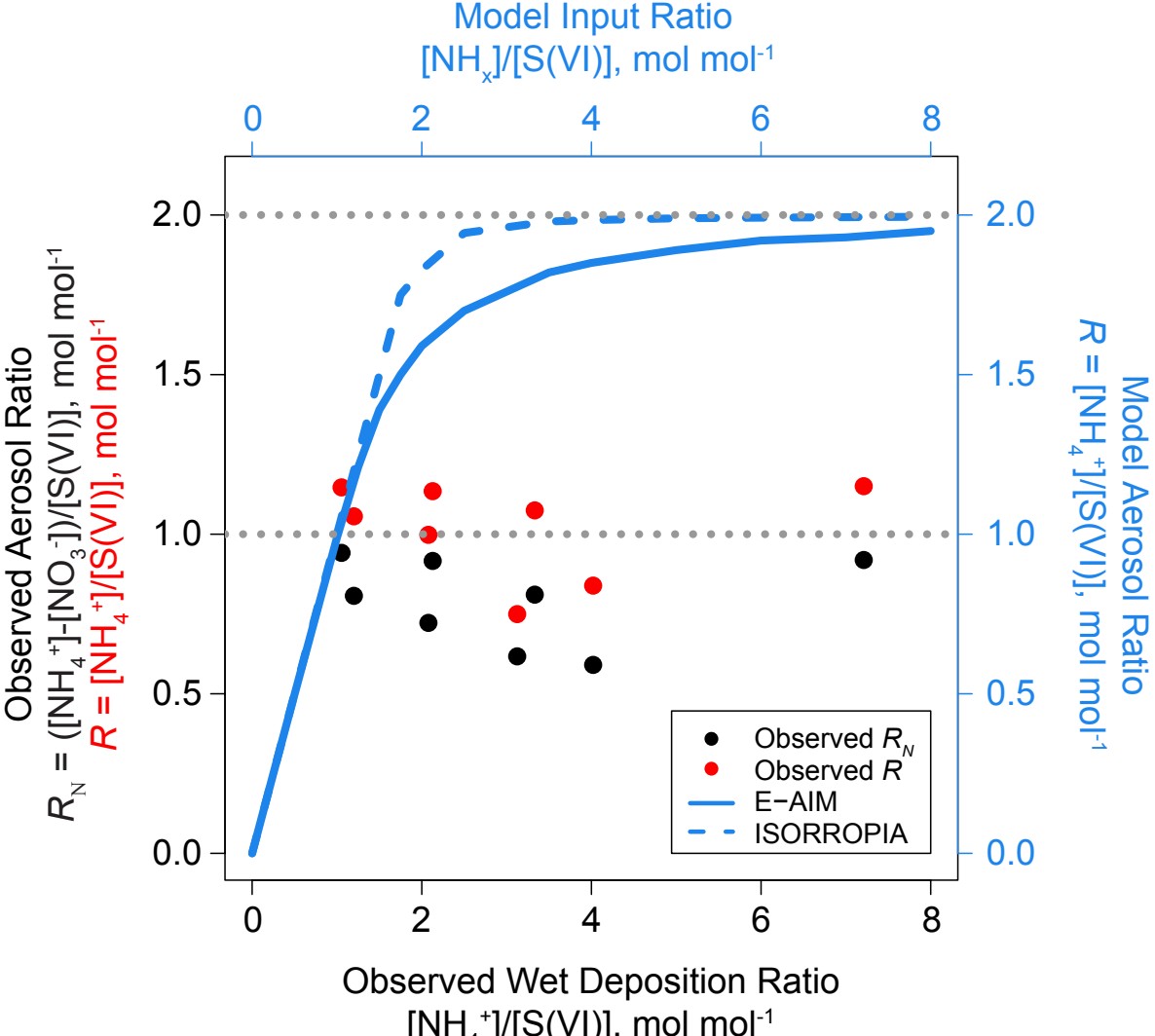

**Figure 3.** Relationship between the ammonium-sulfate ratio in aerosol and in precipitation. The points show the mean observed aerosol ratios from CSN sites vs. the wet deposition flux ratios from NADP sites for summer 2013 at collocated sites in the Southeast US (95-81.5° W, 30.5-37° N) on a 0.5°×0.5° grid. The black points remove ammonium associated with $NH_4NO_3$ and the red points do not. The gray dotted lines show the ratio $R = 1$ corresponding to $NH_4HSO_4$ and $R = 2$ corresponding to $(NH_4)_2SO_4$. The blue curves show the thermodynamic model curves as in Figure 1 but for both E-AIM IV and ISORROPIA II. Both models are applied in the forward mode with total input of $NH_x$ and S(VI) as constraint and for 2 µg m$^{-3}$ S(VI) at 298 K and 70% relative humidity.

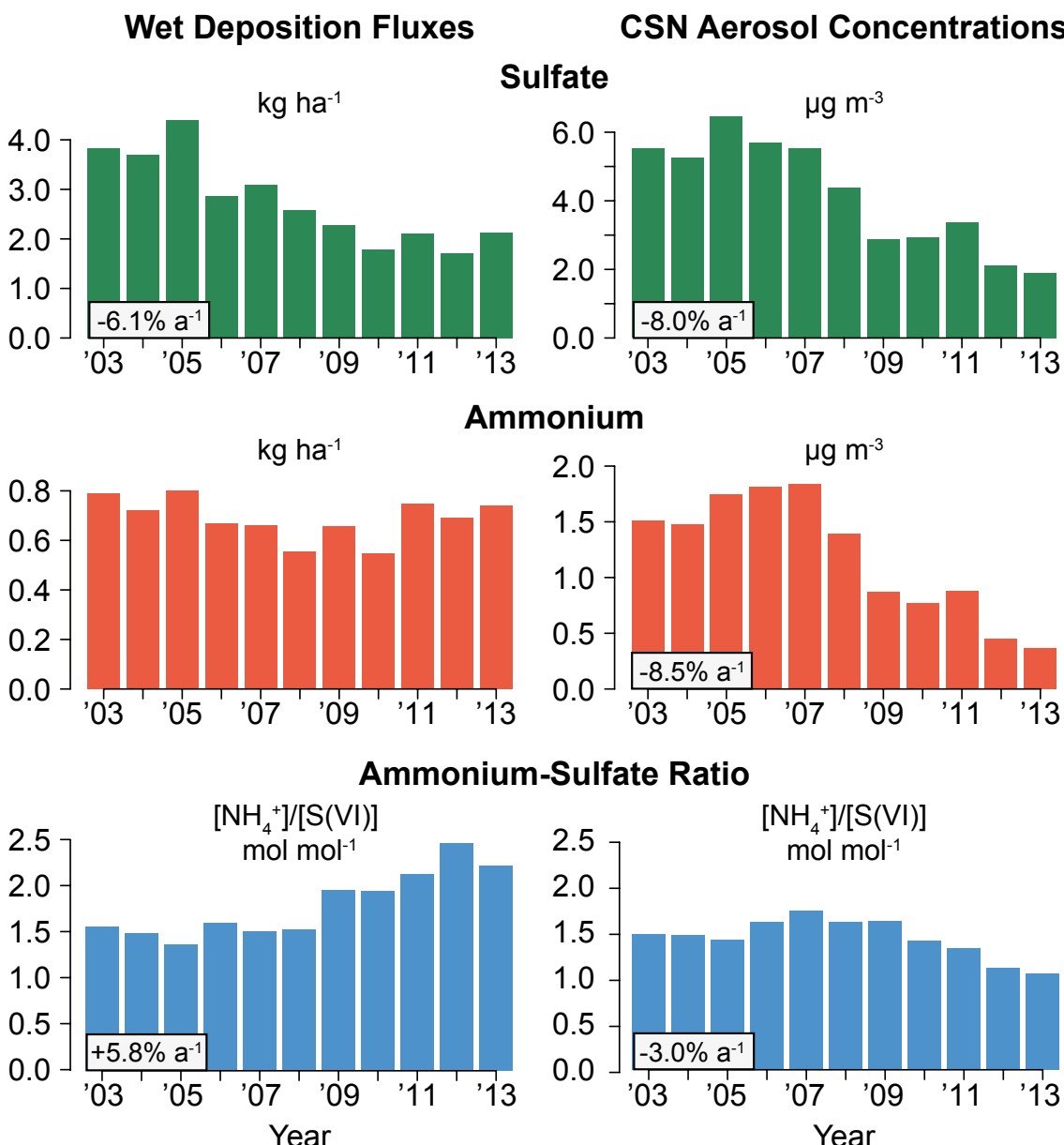

**Figure 4.** 2003-2013 summertime (JJA) trends of sulfate, ammonium, and ammonium-sulfate ratios in wet deposition and aerosol for the Southeast US (95-81.5° W, 30.5-37° N). Values are averages for the NADP and CSN sites in Figure 2. Trends are calculated using the Theil-Sen estimator and are shown when significant at a 95% confidence level.

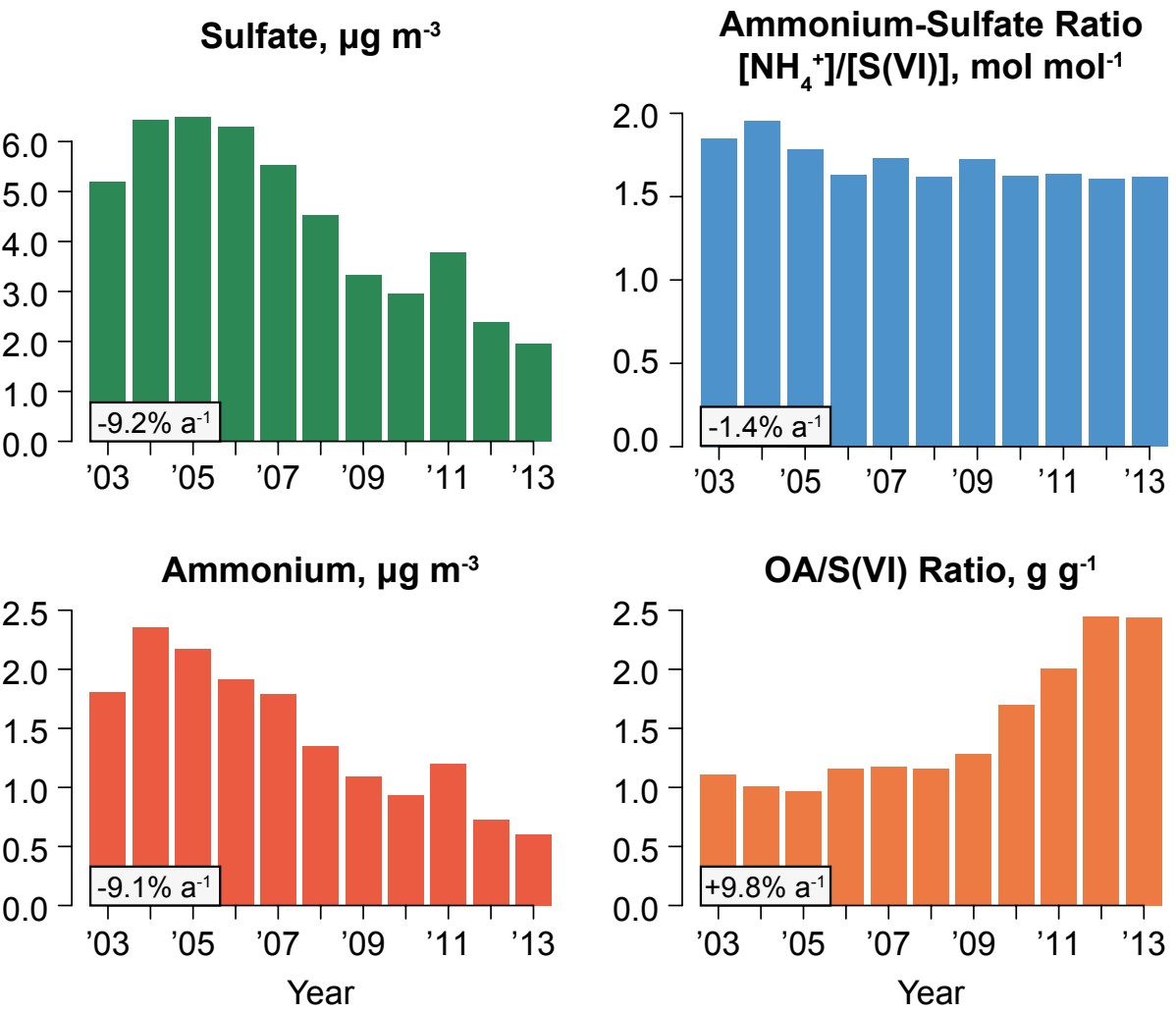

**Figure 5.** 2003-2013 summertime (JJA) trends in aerosol concentrations and ratios at the five SEARCH sites (BHM, CTR, JST, OLF, YRK) with locations shown in Figure 2. The organic aerosol (OA) concentration is inferred from measured organic carbon (OC) and an OA/OC mass ratio of 2.24 (Canagaratna et al., 2015; Kim et al., 2015). Trends are calculated using the Theil-Sen estimator and are shown when significant at a 95% confidence level.

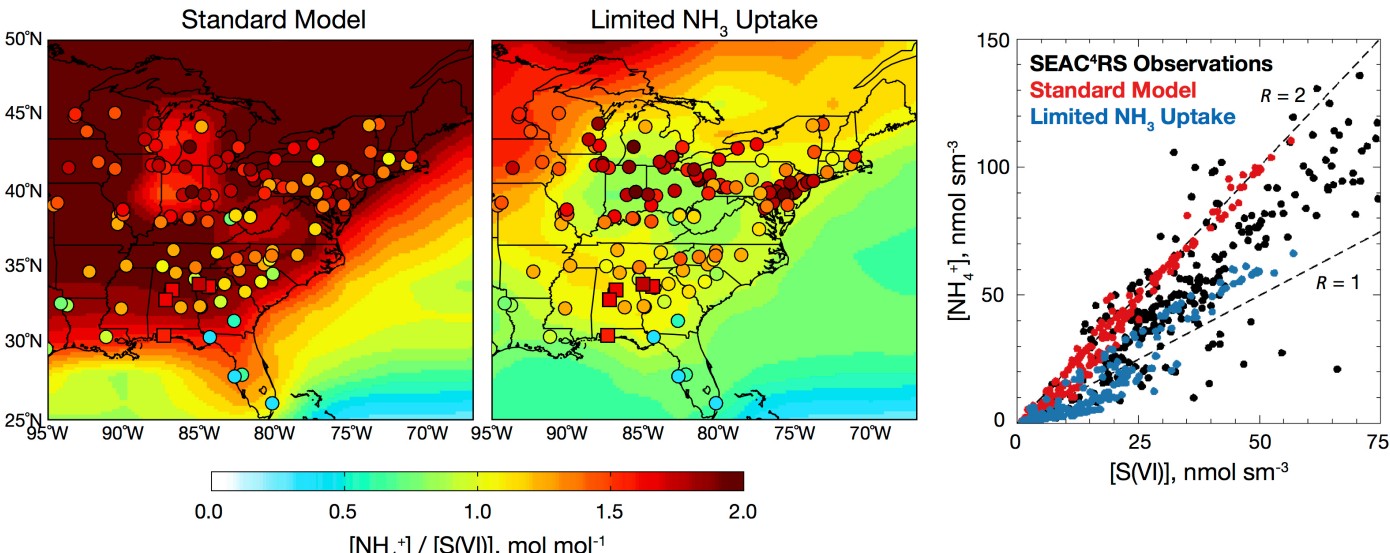

**Figure 6.** Ammonium-sulfate aerosol ratio $R = [NH_4^+]/[S(VI)]$ in the GEOS-Chem chemical transport model and comparison to observations in August 2013. The left and central panels show mean surface air values in the model (background contours) and in the observations at the CSN and SEARCH sites (circles and squares, respectively). The left panel shows results from the standard model assuming sulfate-nitrate-ammonium (SNA) aerosol thermodynamics, while the central panel shows results from the model including kinetic mass transfer limitation to ammonia uptake by SNA aerosol. The right panel compares the two model simulations to aircraft observations over the Southeast US below 2 km altitude from the SEAC$^4$RS aircraft campaign. The model is sampled along the flight tracks (Kim et al., 2015). "sm$^{-3}$" refers to standard cubic meter of air at standard conditions of temperature and pressure (273 K, 1 atm), so that nmol sm$^{-3}$ is a mixing ratio unit. Dashed lines indicate the ratios $R = 1$ corresponding to NH$_4$HSO$_4$ and $R = 2$ corresponding to (NH$_4$)$_2$SO$_4$.

**Figure 7.** Gas-phase concentrations of ammonia at the Southeast US SEARCH sites in summer (JJA) 2013 (Hansen et al., 2003). Values are midday averages (10-16 local time) for the individual SEARCH sites shown in Figures 2 and 6 and for individual days. GEOS-Chem results are shown for the standard model assuming sulfate-nitrate-ammonium (SNA) aerosol thermodynamics and the model including kinetic mass transfer limitation to ammonia uptake by SNA aerosols. Solid lines show reduced major axis regressions and the 1:1 line is dashed. Correlation coefficients ($r$) and regression slopes ($S$) are given inset.