# Peer review of "Inconsistency of ammonium-sulfate aerosol ratios with thermodynamic models in the eastern US: a possible role of organic aerosol"

_Atmospheric Chemistry and Physics, 2016_

## Referee Comment (RC1) · Anonymous Referee #1 · 5 Jul 2016

The crux of the puzzle being addressed in this paper is stated on page 2, lines 15 and 16, which states: [There appears to be a mechanism that maintains the neutralization ratio at a low value even as the SO2/NH3 emission ratio decreases.] Ignoring for the moment the ambiguity of the term neutralization ratio (discussed below), this very issue was identified in a recent paper [Weber et al., 2016] where it was shown that a thermodynamic model predicts this precise drop in neutralization ratio (ie, molar ratio) with decreasing SO4= and steady NH3 levels. The cause was identified to be due to the semi-volatile nature of ammonia and the insensitivity of the system to gas phase NH3 concentrations. Equilibrium partitioning of ammonia predicts lower ammonium-sulfate molar ratios as sulfate concentrations drop. The model was verified by comparison

to observations [Guo et al., 2016; Guo et al., 2015]; the thermodynamic model predictions of the partitioning (gas- particle concentration ratios) of semi-volatile species, such as ammonia and nitric acid, are consistent with observations. Furthermore, the paper explained why little nitrate aerosol has been observed despite large reductions in sulfate.

In this paper, the authors propose a different cause for acidity (actually a NH4+/2SO4= molar ratios less than 1) despite the presence of gas phase ammonia. They suggest that impedance of gas phase ammonia uptake by fine aqueous particles due to an outer organic film is the cause. Simply put, since sulfate is nonvolatile, this increases the time to equilibrium, results in an average (I guess) predicted lower particle ammonium concentration, which lowers the molar ratio. There is no discussion on what happens when equilibrium is reached; presumably this model cannot explain the molar ratios at that point. A question is, why the different conclusions. One explanation may be due to the different analysis approach taken. One group has focused on pH predictions, and then using the predicted pH to assess it's affects on verifiable processes (comparing predicted and measured partitioning of various species), whereas in this work, the question is framed around observed vs predicted ammonium/sulfate molar ratios and using them to interpret ambient data (referred to here as MR). No other verification is provided (ie, comparison of some pH dependent process or species concentration to observations, such as nitrate aerosol).

First off, do the authors have a possible explanation why simply ammonia volatility is insufficient to explain these observations? It would seem prudent to least include a brief discussion stating that an alternative explanation to what they are proposing has been offered, briefly describe it and state why an alterative explanation is necessary. The logic used here seems odd. The authors use E-AIM or ISORROPIA in the analysis (and seem to accept that they largely agree with each other), but adjust the NH3 uptake to get these model predicted MR to agree with observations. But it has been shown that the same model (ISORROPIA) explains these observations. I conclude the issue

Interactive
comment

is that MRs are not a robust nor reliable way to assess model predicted aerosol pH nor it's effects, and this is the fundamental cause for the different views.

Why do the authors use MRs when investigating particle pH? Although commonly used in many different forms (e.g., ion balances, gas ratios, etc) it has been shown that MR do not provide quantitative insight on pH [Guo et al., 2016; Guo et al., 2015; Hennigan et al., 2015; Weber et al., 2016], but instead can lead to confusion and imprecise conclusions (see a list of them in specific comments below). Given this, it is not clear what their utility is. The author's show that the particle MR is not related to pH in any straightforward way (see fig 1) and that pH remains very low despite MR near or approaching 1. Given that pH is what controls aerosol processes related to acidity, such as partitioning of semi-volatile acids, acid catalyzed reactions, etc, and not MR, the use of MR to infer acidity is problematic.

As an example, what do the authors mean by their commonly used term, sulfate neutralization? Or sometimes, the term is just, aerosol neutralization, such as in the figures and in the text (pg 5 line 32). It is also in the title of the paper. Strictly, is one to interpret this to mean that the MR is 1 or greater? However, it also seems to be used to imply (or at least it gives the impression) that the authors are referring to overall particle acidity (i.e., pH), such that a MR of 1 implies a neutral aerosol. Example, page 8 line 7 and 8 states: quote [The aerosol in the standard model is fully neutralized throughout the eastern US (mean f = 1.00), at odds with observations.], end quote. But than Fig 1 shows that the pH is nowhere near neutral as f approaches 1; a contradiction. Including with their Fig 1, a plot of specifically f vs pH, would be very insightful, I believe. There are many instances in this paper of similar ambiguous statements (discussed more below).

In any case, molar ratios are used in this work. The author's point is that the molar ratio predicted by the thermodynamic model is higher than what is observed, and so there is either an issue with the thermodynamic model or with the assumptions the model is based on. This paper proposes the latter, that NH3 mass transport limitations due to

organic films are the cause and that it takes over a day for equilibrium to be established, thus the strict use of thermodynamic models, which assume equilibrium, give incorrect results. However, maybe other causes for the discrepancy are possible, eg, relating to oversimplification when using the model, such as lack of size dependent composition and the mixing state of the aerosol. What if one simply adds a small amount of other non-volatile cations, such as mineral dust, to the model; a reasonable thing to do since the data are PM2.5. This should lower the predicted MRs. Considering the likely small amount of other cations needed to significantly alter the MR, and measurement uncertainties (discussed more below), it is not clear how the authors can outright concluded that other cations play no role (ie, reference to Kim 2015 and Liao 2015). More details, provided in this paper, such as including sensitivity tests with various amounts of other cations could be very insightful.

Another issue that is not clear is the extensive comparisons of parameters, such as aerosol composition and MRs, between measurements of wet deposition versus fine particles. In the abstract it is used to infer that gas phase ammonia is in excess (meaning, presumably, not equal to zero). Fig 3 compares wet deposition neutralization ratios to those of fine particles. Generally the authors find that the wet deposition trends for sulfate and ammonium follow SO2 and NH3 emission trends and that the neutralization ratios (ie, MRs) are close to 1, as expected for excess NH3, whereas neither is true for fine particles. Are the authors inferring this provides evidence for some additional process affecting the fine particles? It is actually thermodynamically consistent and is due to the vastly different concentrations of liquid water (many orders of magnitude) associated with these systems. It is not clear how differences in MRs between fine haze particles and fog/cloud particles support the hypothesis of this manuscript and needs to be clarified? Maybe that was not the intent, but the use of wet deposition is confusing and should be clarified.

Overall, this paper proposes an explanation for the persistent particle acidity despite reductions in SO2 emissions in the eastern US. Understanding the cause is clearly

of great importance. In my view, the proposed explanation is speculative and largely unsupported because the use of MRs in their analysis is highly suspect. Furthermore, a much simpler explanation exists. I do believe the authors could test their hypothesis further; say by predicting concentrations of semi-volatile aerosol components that depend on pH, in different settings. I am not against publishing this paper, but believe that significant clarifications are needed throughout prior to publication. More specific detailed questions are provided below.

Specific comments (in the following [ . . . ] indicates a quote from the manuscript)

How was the pH in Fig 1 calculated? Is it the equilibrium predicted pH, (ie, no kinetic limitation due to organic layer), or avg pH in the overall system that includes predicted pH before equilibrium is reached?

It is often not clear exactly what data were used for gas phase ammonia in the thermodynamic models. Was the wet deposition data used under the assumption that it represents the total sulfur and ammonia, and thus as model input? This is likely reasonable, but leads to much higher uncertainty than using direct measurements of the species involved (sulfate, ammonium, ammonia). Granted the wet deposition data is more widely available then NH3, but still analysis for studies where all the data exists would greatly support the hypothesis of this work.

If this organic film inhibits uptake of NH3, how it is likely to influence other semi-volatile species (nitric acid, HCl) and how will this impact the partitioning of these compounds. Will the model be able to accurately predict partitioning of these other species? Why not use this as a means of testing the hypothesis?

One might expect this proposed process to be highly important in biomass burning plumes where the organic aerosol fraction is very high. Thermodynamic models appear to also work well in these cases [Bougiatioti et al., 2016]. Based on the arguments in this work one may expect them not to. The authors could investigate if there is other published work on thermodynamic predictions in smoke plumes that support their

hypothesis.

In the Abstract it states: [. . .sulfate aerosol is not fully neutralized even in the presence of excess ammonia, at odds with thermodynamic equilibrium models]. Is it ever shown in this paper that this statement is true? There is only a general reference to Seinfeld and Pandis. A thermodynamic calculation that supports this statement is needed. Also from the Abstract, last line states: [If sulfate aerosol becomes more acidic as OA/sulfate ratios increase, then controlling SO2 emissions to decrease sulfate aerosol will not have the co-benefit of suppressing acid-catalyzed secondary organic aerosol (SOA) formation.] This requires knowing how this proposed process specifically affects pH, which I don't believe is ever discussed. That is, what is the effect of this organic coating on aerosol pH?

Page 3, line 2-6 states: [Nitrate partitions into the aerosol only when ammonia is in excess of sulfate neutralization, limited by the supply of excess ammonia and dependent on temperature.] Can one provide a thermodynamic analysis proving this is true? Fig A above, suggests otherwise.

Page 3 last sentence states: [In any case, sufficient ammonia is emitted in the eastern US to fully neutralize the sulfate contributed by SO2 emissions.] Again, can this statement be shown to be true based on a thermodynamic calculation?

Page 4 it states: [ alkaline cations other than ammonium do not contribute significantly to the neutralization (Kim et al., 2015),...] A few lines explicitly stating how this is shown in Kim would be very useful (as noted above).

Page 4 gives f(N) from various studies, including AMS data from SOAS and SEAC4RS. Is the reported +/- the standard deviation of the data or the uncertainty in f(N). If not uncertainty, what is the uncertainty? (As an example, AMS uncertainty of 35% is often reported, which would lead to an uncertainty of about 50% for f(N). Also, in side by side comparisons, such as in SOAS and WINTER studies, systematic differences in AMS and PILS measurements of sulfate, nitrate and ammonium can easily by 20%. I

suspect similar levels apply to network data. It seems that measurement uncertainty alone, both random and systematic (the latter more important here), makes asserting that there is a difference between measured and model predicted f(N) tenuous.

Page 5 Lines 29 to 31 it states: [We see that the observed aerosol neutralization does not follow thermodynamic predictions. . .] A similar statement is made in the last line on page 5. Specifically, what thermodynamic predictions are being referred to? Was a full thermodynamic model run to verify this? Please provide specific evidence to support this statement.

Page 6 it states: [. . .the sulfate aerosol is far from neutralized, and one would expect under these conditions that the aerosol neutralization ratio should increase as the supply of sulfate decreases. But this is not observed.] What does, [one would expect], mean? Again, can these statements be proven with a thermodynamic calculation?

The proposed mechanism to bring model predictions closer to observed MRs is to lower the NH3 uptake. This results in the equilibrium time scales increasing to roughly a day or more. Presumably this will also affect fine particle nitric acid equilibrium time scales. Even ignoring any effect this film may have on nitric acid uptake. This could have a large impact on predicted fine particle nitrate concentrations, given that in many locations there is a significant mass of cations in the coarse mode (Na+, Ca2+, . . .) that can form non-volatile forms of nitrate, if given sufficient time. Thus, it would seem the viability of this proposed mechanism could be tested by predicting nitrate aerosol levels and comparing to observations, for example during cooler periods when more nitrate partitions to fine particles. It has been noted in a number of publications that ISORROPIA can accurately predict nitrate (without this proposed mechanism). How do the authors explain this if by their mechanism the aerosol is not in equilibrium?

Bougiatioti, A., P. Nikolaou, I. Stavroulas, G. Kouvarakis, R. Weber, A. Nenes, M. Kanakidou, and N. Mihalopoulos (2016), Particle water and pH in the eastern Mediterranean: Sources variability and implications for nutrients availability, Atm. Chem.

[Figure]

Phys., 16, 4579-4591.

Guo, H., A. P. Sullivan, P. Campuzano-Jost, J. C. Schroder, F. D. Lopez-Hilfiker, J. E. Dibb, J. L. Jimenez, J. A. Thornton, S. S. Brown, A. Nenes, and R. J. Weber (2016), Fine particle pH and the partitioning of nitric acid during winter in the northeastern United States, J. Geophys. Res., in review.

Guo, H., L. Xu, A. Bougiatioti, K. M. Cerully, S. L. Capps, J. R. Hite, A. G. Carlton, S.-H. Lee, M. H. Bergin, N. L. Ng, A. Nenes, and R. J. Weber (2015), Predicting particle water and pH in the southeastern United States, Atmos. Chem. Phys., 15, 5211–5228.

Hennigan, C. J., J. Izumi, A. P. Sullivan, R. J. Weber, and A. Nenes (2015), A critical evaluation of proxy methods used to estimate the acidity of atmospheric particles, Atm. Chem. Phys., 15, 1-15.

Weber, R. J., H. Guo, A. G. Russell, and A. Nenes (2016), High aerosol acidity despite declining atmospheric sulfate concentrations over the past 15 years, Nature Geoscience, 9(10.1038/ngeo2665), 282-285.

---

## Author Comment (AC1) · 8 Jul 2016

Thank you for the detailed review. It appears there is some confusion about the constraints used in our thermodynamic calculations. We use the total ammonia to sulfate ratio (NHx/S(VI)) as the predictor variable to identify a departure from thermodynamic equilibrium in the observations, not the ammonium to sulfate ratio (which the reviewer refers to as MR). Additionally, our paper does not focus on aerosol pH but on the aerosol neutralization ratio, and the discussion of aerosol pH at the beginning of the paper is only to make the point that the aerosol is acidic even if ammonia is in large excess. As the reviewer points out, this simply comes out of the thermodynamics but we expect that some readers may be confused. We will clarify these issues and provide a

point-by-point response to this review at the end of the discussion period. We may not be able to convince the reviewer but ultimately look forward to the continuation of this discussion of aerosol thermodynamics in the open literature.

---

## Referee Comment (RC2) · Anonymous Referee #2 · 16 Jul 2016

The authors present data suggesting that the sulfate in the Eastern US is not fully neutralized by ammonia despite the availability of the latter in the gas phase and argue that this could be due to mass transfer delays between the gas and particulate phase caused by organics. This is clearly an important issue that has received a lot of attention in the literature during the last few years. The paper is well written, however, the evidence provided to support its major conclusion is rather weak and it neglects other simpler explanations. It also neglects important work that has been published recently explaining similar observations with existing aerosol thermodynamics models. These problems are discussed below.

(1) Evidence that the aerosol is in equilibrium in the Southeast US The authors conclude from their data analysis that the aerosol in the eastern US is not in equilibrium probably due to limitations of the mass transfer of ammonia due to the organics. However, they neglect at least two studies that have used high quality gas-phase ammonia measurements in the same area and season to investigate this issue and reached the opposite conclusion. Weber et al. (2016) showed that the measured gas-phase ammonia was in equilibrium with the particulate phase during SOAS (see for example Figure 1 in that paper). They used 1-hr as averaging time therefore the equilibration timescale should be much less than this. If there was a significant delay in the mass transfer of ammonia between the two phases there should be significant observed discrepancies. Similar conclusions were reached by Nowak et al. (2006) for the 2002 ANARChE study in Atlanta. They concluded that the agreement between the measured and predicted ammonia within the uncertainties suggests that the assumption of thermodynamic equilibrium on the 7.5 min timescale is appropriate for most of the ANARChE data examined here.

(2) Alternative explanations Weber et al. (2016) have argued that the incomplete neutralization of sulfate in the southeast US and the corresponding changes during the last 15 years are in general consistent with our current understanding of inorganic aerosol thermodynamics. They did not need to invoke delays in mass transfer and lack of equilibrium between the two phases. The authors appear to try something similar in Figure 3 but in a rather convoluted way using wet deposition data (please see comment 6 below).

(3) Model evaluation The use of GEOS-Chem to test the hypothesis of delays in the mass transfer of ammonia is the most original aspect of the current work. However, the evaluation of the corresponding GEOS-Chem predictions is rather superficial given the availability of the SOAS measurements during the same summer. The ability of GEOS-Chem to reproduce observed gas-phase ammonia concentrations is critical for the authors' argument. For example, the problems of the base case simulation could be due to the overprediction of ammonia availability (e.g., due to errors in ammonia

emissions) and the assumed delay in mass transfer could be correcting one error by introducing another.

(4) Terminology and details about calculated quantities I found the terminology used in the paper quite confusing. For example, the terms "sulfate neutralization" and "aerosol neutralization" are used throughout the paper instead of the ammonium to sulfate molar or equivalent ratio. Sometimes the nitrate is also used sometimes it is not. In all cases the other cations that could, in principle at least, be neutralizing sulfate are not included in this neutralization ratio.

It is not clear if the different ratios shown throughout the paper correspond to PM1, PM2.5, PM10 or something else. This detail is critical given the importance of calcium, magnesium, etc., for the coarse particles.

It is also not clear how the authors estimate the averages of the different ratios. Do they average the concentrations and then estimate the ratios or do they estimate the ratios with some averaging time (daily?) and then average them?

(5) Originality of hypothesis The same hypothesis regarding the organic aerosol role in mass transfer of ammonia in the southeast US has been presented Kim et al. (ACP, 2015) with a number of common authors in the two studies. This should be discussed in the introduction of the paper.

(6) Use of wet deposition data The authors use wet deposition measurements as practically a surrogate of total (gas and particulate) ammonia. This is rather tricky given that rainfall takes place during specific meteorological conditions, clouds can produce sulfate, and there are of course different wet removal efficiencies for ammonia and sulfate. Despite these problems, the paper does not address the potential biases that could be introduced in the analysis because of the use of these data. This could be one of the reasons for the differences between the conclusions here and those of other studies. I realize that gas-phase ammonia measurements exist only during specific field campaigns, but there are enough of them available both in the US and Europe.

Use of these measurements is clearly preferable to the wet deposition data.

(7) Role of organosulfates and crustal elements Some additional information is needed regarding the potential role of organosulfates and crustal elements for the present analysis. The authors appear to assume that they are negligible for the purposes of this work. However, some quantitative arguments are needed taking for example advantage of the SOAS measurements.

(8) Some additional minor points:

Page 2, lines 2-4. This is clearly not true when there are other cations present.

Figure 2a is rather misleading given that most of the sulfur dioxide is emitted by point sources. It should probably be replaced with a table with the emissions in different regions of the US (e.g., southeast, northeast, etc.).

---

## Author Comment (AC2) · 26 Sep 2016

We thank the two anonymous reviewers for their helpful comments. The reviewer comments are in black and our responses are in blue and include page and line numbers where changes were made to the accompanying manuscript.

**Response to Reviewer #1**

The crux of the puzzle being addressed in this paper is stated on page 2, lines 15 and 16, which states: [There appears to be a mechanism that maintains the neutralization ratio at a low value even as the SO2/NH3 emission ratio decreases.] Ignoring for the moment the ambiguity of the term neutralization ratio (discussed below), this very issue was identified in a recent paper [Weber et al., 2016] where it was shown that a thermodynamic model predicts this precise drop in neutralization ratio (ie, molar ratio) with decreasing SO4= and steady NH3 levels. The cause was identified to be due to the semi-volatile nature of ammonia and the insensitivity of the system to gas phase NH3 concentrations. Equilibrium partitioning of ammonia predicts lower ammonium-sulfate molar ratios as sulfate concentrations drop. The model was verified by comparison to observations [Guo et al., 2016; Guo et al., 2015]; the thermodynamic model predictions of the partitioning (gas- particle concentration ratios) of semi-volatile species, such as ammonia and nitric acid, are consistent with observations. Furthermore, the paper explained why little nitrate aerosol has been observed despite large reductions in sulfate.

We have amended the text (page 2 line 24-page 3 line 1) to address the findings of Weber et al. (2016) and to put our work in context of that paper. This is very useful. Our work goes beyond the Weber paper by pointing out the thermodynamic inconsistency in the simultaneous observations of low ammonium-sulfate ratios ($R$) and significant gas-phase ammonia. The Weber paper claims consistency but the values of $R$ from their model calculations (Figure 2, "past") are higher than observed. We now point out that low values of nitrate are consistent with low values of $R$, citing Guo et al. (2015) and Weber et al. (2016) (page 6 lines 2-3).

In this paper, the authors propose a different cause for acidity (actually a NH4+/2SO4= molar ratios less than 1) despite the presence of gas phase ammonia. They suggest that impedance of gas phase ammonia uptake by fine aqueous particles due to an outer organic film is the cause. Simply put, since sulfate is nonvolatile, this increases the time to equilibrium, results in an average (I guess) predicted lower particle ammonium concentration, which lowers the molar ratio. There is no discussion on what happens when equilibrium is reached; presumably this model cannot explain the molar ratios at that point. A question is, why the different conclusions. One explanation may be due to the different analysis approach taken. One group has focused on pH predictions, and then using the predicted pH to assess it's affects on verifiable processes (comparing predicted and measured partitioning of various species), whereas in this work, the question is framed around observed vs predicted ammonium/sulfate molar ratios and using them to interpret ambient data (referred to here as MR). No other verification is provided (ie, comparison of some pH dependent process or species concentration to observations, such as nitrate aerosol).

Our main focus in this paper is to point out the thermodynamic inconsistency between observed low ammonium-sulfate ratios and gas-phase ammonia, clarified now in the text (page 2 line 32-

page 3 line 1). We would be remiss if we didn't come up with a tentative explanation, but we now use the word "tentative" in the abstract and text, because it is certainly not our intention to claim that this crude parameterization of kinetic limitation to uptake is the final word. We point out in the conclusions that this explanation would have a number of other implications but to our knowledge there are no available observations that would rule it out. As requested by the reviewer we have added an evaluation of gas-phase ammonia (page 9 lines 18-22, 29-32 with a new Figure 7, page 24).

First off, do the authors have a possible explanation why simply ammonia volatility is insufficient to explain these observations? It would seem prudent to least include a brief discussion stating that an alternative explanation to what they are proposing has been offered, briefly describe it and state why an alterative explanation is necessary. The logic used here seems odd. The authors use E-AIM or ISORROPIA in the analysis (and seem to accept that they largely agree with each other), but adjust the NH3 uptake to get these model predicted MR to agree with observations. But it has been shown that the same model (ISORROPIA) explains these observations. I conclude the issue is that MRs are not a robust nor reliable way to assess model predicted aerosol pH nor it's effects, and this is the fundamental cause for the different views.

The Weber et al. (2016) paper does not show agreement between model and observed $R$. We now devote a new paragraph of the introduction to point this out – this is indeed important to show how our paper goes beyond Weber et al. so we are very thankful to the reviewer for raising the issue and we believe that we have addressed it in the revision.

Why do the authors use MRs when investigating particle pH? Although commonly used in many different forms (e.g., ion balances, gas ratios, etc) it has been shown that MR do not provide quantitative insight on pH [Guo et al., 2016; Guo et al., 2015; Hennigan et al., 2015; Weber et al., 2016], but instead can lead to confusion and imprecise conclusions (see a list of them in specific comments below). Given this, it is not clear what their utility is. The author's show that the particle MR is not related to pH in any straightforward way (see fig 1) and that pH remains very low despite MR near or approaching 1. Given that pH is what controls aerosol processes related to acidity, such as partitioning of semi-volatile acids, acid catalyzed reactions, etc, and not MR, the use of MR to infer acidity is problematic.

Our focus is not on investigating particle pH. Our focus is on reconciling the joint occurrence of low $R$ and high gas-phase ammonia. We have clarified this in revision, and also completely purged the word "neutralization" from the text at the reviewer's suggestion and to make this clearer.

As an example, what do the authors mean by their commonly used term, sulfate neutralization? Or sometimes, the term is just, aerosol neutralization, such as in the figures and in the text (pg 5 line 32). It is also in the title of the paper. Strictly, is one to interpret this to mean that the MR is 1 or greater? However, it also seems to be used to imply (or at least it gives the impression) that the authors are referring to overall particle acidity (i.e., pH), such that a MR of 1 implies a neutral aerosol. Example, page 8 line 7 and 8 states: quote [The aerosol in the standard model is fully neutralized throughout the eastern US (mean f = 1.00), at odds with observations.], end quote. But than Fig 1 shows that the pH is nowhere near neutral as f approaches 1; a

contradiction. Including with their Fig 1, a plot of specifically f vs pH, would be very insightful, I believe. There are many instances in this paper of similar ambiguous statements (discussed more below).

In order to address the reviewer's concern we have (1) eliminated the "neutralization" terminology throughout the text, and (2) replaced *f* by R = [NH$_4^+$]/[S(VI)] to facilitate comparison with Weber et al. (1) should help to avoid the impression that we are focusing on aerosol pH, which we are not. See response above regarding distinction between pH and MR. Figure 1 (page 18) shows a plot of *R* and pH.

In any case, molar ratios are used in this work. The author's point is that the molar ratio predicted by the thermodynamic model is higher than what is observed, and so there is either an issue with the thermodynamic model or with the assumptions the model is based on. This paper proposes the latter, that NH3 mass transport limitations due to organic films are the cause and that it takes over a day for equilibrium to be established, thus the strict use of thermodynamic models, which assume equilibrium, give incorrect results. However, maybe other causes for the discrepancy are possible, eg, relating to oversimplification when using the model, such as lack of size dependent composition and the mixing state of the aerosol. What if one simply adds a small amount of other non-volatile cations, such as mineral dust, to the model; a reasonable thing to do since the data are PM2.5. This should lower the predicted MRs. Considering the likely small amount of other cations needed to significantly alter the MR, and measurement uncertainties (discussed more below), it is not clear how the authors can outright concluded that other cations play no role (ie, reference to Kim 2015 and Liao 2015). More details, provided in this paper, such as including sensitivity tests with various amounts of other cations could be very insightful.

We now address the lack of size dependent composition in the text (page 4 lines 6-7). We cannot conjure a scenario where size-dependent composition would explain the low value of *R* so it really shouldn't matter to the argument, and indeed Weber et al. didn't bother with it. At the reviewer's suggestion we now specifically quantify the small effect of soil dust cations in the data that we analyze and point out that the effect is small (page 5 lines 25-26).

Another issue that is not clear is the extensive comparisons of parameters, such as aerosol composition and MRs, between measurements of wet deposition versus fine particles. In the abstract it is used to infer that gas phase ammonia is in excess (meaning, presumably, not equal to zero). Fig 3 compares wet deposition neutralization ratios to those of fine particles. Generally the authors find that the wet deposition trends for sulfate and ammonium follow SO2 and NH3 emission trends and that the neutralization ratios (ie, MRs) are close to 1, as expected for excess NH3, whereas neither is true for fine particles. Are the authors inferring this provides evidence for some additional process affecting the fine particles? It is actually thermodynamically consistent and is due to the vastly different concentrations of liquid water (many orders of magnitude) associated with these systems. It is not clear how differences in MRs between fine haze particles and fog/cloud particles support the hypothesis of this manuscript and needs to be clarified? Maybe that was not the intent, but the use of wet deposition is confusing and should be clarified.

We have clarified our use of wet deposition data as it relates to emissions of ammonia and $SO_2$ on page 5 lines 3-6. We have also now included explicit discussion of gas-phase ammonia (page 4 lines 4-6; page 9 lines 18-22, 29-32; page 18, Figure 1; page 24, Figure 7).

Overall, this paper proposes an explanation for the persistent particle acidity despite reductions in SO2 emissions in the eastern US. Understanding the cause is clearly of great importance. In my view, the proposed explanation is speculative and largely unsupported because the use of MRs in their analysis is highly suspect. Furthermore, a much simpler explanation exists. I do believe the authors could test their hypothesis further; say by predicting concentrations of semi-volatile aerosol components that depend on pH, in different settings. I am not against publishing this paper, but believe that significant clarifications are needed throughout prior to publication. More specific detailed questions are provided below
Specific comments (in the following [ . . . ] indicates a quote from the manuscript)

How was the pH in Fig 1 calculated? Is it the equilibrium predicted pH, (ie, no kinetic limitation due to organic layer), or avg pH in the overall system that includes predicted pH before equilibrium is reached?

We added a clarification to the text that the pH in Figure 1 is for thermodynamic equilibrium (page 3, lines 31-32).

It is often not clear exactly what data were used for gas phase ammonia in the thermodynamic models. Was the wet deposition data used under the assumption that it represents the total sulfur and ammonia, and thus as model input? This is likely reasonable, but leads to much higher uncertainty than using direct measurements of the species involved (sulfate, ammonium, ammonia). Granted the wet deposition data is more widely available then NH3, but still analysis for studies where all the data exists would greatly support the hypothesis of this work.

Yes indeed, and we thank the reviewer for the suggestion. We have added an evaluation of gas-phase ammonia measurements compared to model predictions (page 9 lines 18-22, 29-32; Figure 7 page 24) as well as additional discussion of ammonia predicted by thermodynamic models (page 4 lines 4-6; page 18, Figure 1). The wet deposition data shown on page 20, Figure 3 is compared to thermodynamic models under the assumption that it represents total S(VI) and $NH_x$ and is now stated in the text (page 7 lines 5-6).

If this organic film inhibits uptake of NH3, how it is likely to influence other semi-volatile species (nitric acid, HCl) and how will this impact the partitioning of these compounds. Will the model be able to accurately predict partitioning of these other species? Why not use this as a means of testing the hypothesis?

We now stress in the text that the hypothesis is tentative and that our representation of the OA effect is crude. Our focus here is mostly to offer a direction for future work.

One might expect this proposed process to be highly important in biomass burning plumes where the organic aerosol fraction is very high. Thermodynamic models appear to also work well in these cases [Bougiatioti et al., 2016]. Based on the arguments in this work one may expect them

not to. The authors could investigate if there is other published work on thermodynamic predictions in smoke plumes that support their hypothesis.

Bougiatioti et al. (2016) focused on predicting aerosol pH and on average conditions in that study were more sulfate-rich than is typical of the Southeast US summer (33.8% organic aerosol by mass in Bougiatioti et al., 2016 in contrast to 55% organic aerosol in August-September 2013 in the Southeast US shown in Kim et al., 2015).

In the Abstract it states: [. . .sulfate aerosol is not fully neutralized even in the presence of excess ammonia, at odds with thermodynamic equilibrium models]. Is it ever shown in this paper that this statement is true? There is only a general reference to Seinfeld and Pandis. A thermodynamic calculation that supports this statement is needed. Also from the Abstract, last line states: [If sulfate aerosol becomes more acidic as OA/sulfate ratios increase, then controlling SO2 emissions to decrease sulfate aerosol will not have the co-benefit of suppressing acid-catalyzed secondary organic aerosol (SOA) formation.] This requires knowing how this proposed process specifically affects pH, which I don't believe is ever discussed. That is, what is the effect of this organic coating on aerosol pH?

Figure 1 (page 18) shows that ISORROPIA predicts $R$ approaching 2 in the presence of excess ammonia and is stated in the text (page 4 lines 2-4).

We now state in the Conclusions (page 10 lines 23-25) the implication of the organic coating on the aerosol pH trend and how it could explain the observations of Weber et al. (2016).

Page 3, line 2-6 states: [Nitrate partitions into the aerosol only when ammonia is in excess of sulfate neutralization, limited by the supply of excess ammonia and dependent on temperature.] Can one provide a thermodynamic analysis proving this is true? Fig A above, suggests otherwise.

Figure 1 does not include nitrate, stated in the text (page 3 lines 23-24). Citations of the relevant literature supporting the statement in the text have been added (page 3 lines 21-22; Ansari and Pandis, 1998; Park et al., 2004).

Page 3 last sentence states: [In any case, sufficient ammonia is emitted in the eastern US to fully neutralize the sulfate contributed by SO2 emissions.] Again, can this statement be shown to be true based on a thermodynamic calculation?

This is a statement based on the relative emissions of $NH_3$ to $SO_2$ shown in Figure 2 (page 19) and has been clarified in the text (page 4 lines 28-29).

Page 4 it states: [ alkaline cations other than ammonium do not contribute significantly to the neutralization (Kim et al., 2015),...] A few lines explicitly stating how this is shown in Kim would be very useful (as noted above).

We have added discussion of the influence of alkaline cations in the text (page 5 lines 22-26).

Page 4 gives f(N) from various studies, including AMS data from SOAS and SEAC4RS. Is the reported +/- the standard deviation of the data or the uncertainty in f(N). If not uncertainty, what is the uncertainty? (As an example, AMS uncertainty of 35% is often reported, which would lead to an uncertainty of about 50% for f(N). Also, in side by side comparisons, such as in SOAS and WINTER studies, systematic differences in AMS and PILS measurements of sulfate, nitrate and ammonium can easily by 20%. I suspect similar levels apply to network data. It seems that measurement uncertainty alone, both random and systematic (the latter more important here), makes asserting that there is a difference between measured and model predicted f(N) tenuous.

The mean values of $R$ stated in this manuscript are provided with a propagated uncertainty and we now clarify this in the text (page 5 lines 28-29). SEARCH $PM_{2.5}$ measurements have a $1\sigma$ precision of 3% for sulfate and 5% for ammonium (as derived from the precision statistics reported by Edgerton et al., 2005) and CSN measurements have a $1\sigma$ precision of 6% for sulfate and 8% for ammonium (Flanagan et al. 2006). We have added to the text the corresponding propagated uncertainties for $R$ (page 6 lines 8-9). We also note that while AMS absolute uncertainties of 35% for inorganic species are reported, the uncertainties on ratios of AMS species are lower, since the dominant source of uncertainty (collection efficiency) cancels out in the species ratio.

Page 5 Lines 29 to 31 it states: [We see that the observed aerosol neutralization does not follow thermodynamic predictions. . .] A similar statement is made in the last line on page 5. Specifically, what thermodynamic predictions are being referred to? Was a full thermodynamic model run to verify this? Please provide specific evidence to support this statement.

These statements refer to Figure 3 (page 20) as described in the text (page 7 lines 3-8) in which two thermodynamic models (E-AIM and ISORROPIA) are run and compared to observations. We have eliminated the term "neutralization" throughout the paper at the reviewer's suggestion and to clarify our intent.

Page 6 it states: [. . .the sulfate aerosol is far from neutralized, and one would expect under these conditions that the aerosol neutralization ratio should increase as the supply of sulfate decreases. But this is not observed.] What does, [one would expect], mean? Again, can these statements be proven with a thermodynamic calculation?

The statement refers to the thermodynamic predictions in Figures 1 and 3 that show the ammonium-sulfate ratio should increase as the $NH_x/S(VI)$ ratio increases and the text has been clarified on page 7 lines 17-22. Again, getting rid of "neutralization" should also help.

The proposed mechanism to bring model predictions closer to observed MRs is to lower the NH3 uptake. This results in the equilibrium time scales increasing to roughly a day or more. Presumably this will also affect fine particle nitric acid equilibrium time scales. Even ignoring any effect this film may have on nitric acid uptake. This could have a large impact on predicted fine particle nitrate concentrations, given that in many locations there is a significant mass of cations in the coarse mode (Na+, Ca2+, . . .) that can form non-volatile forms of nitrate, if given sufficient time. Thus, it would seem the viability of this proposed mechanism could be tested by predicting nitrate aerosol levels and comparing to observations, for example during cooler

periods when more nitrate partitions to fine particles. It has been noted in a number of publications that ISORROPIA can accurately predict nitrate (without this proposed mechanism). How do the authors explain this if by their mechanism the aerosol is not in equilibrium?

Nitrate concentrations are very low in the Southeast US summer, now stated in the text (page 5 lines 19-22). We have added an evaluation of gas-phase ammonia in GEOS-Chem compared to observations from the SEARCH network (page 9 lines 18-22, 29-32; Figure 7, page 24) as further supporting evidence of a kinetic limitation.

Bougiatioti, A., P. Nikolaou, I. Stavroulas, G. Kouvarakis, R. Weber, A. Nenes, M. Kanakidou, and N. Mihalopoulos (2016), Particle water and pH in the eastern Mediterranean: Sources variability and implications for nutrients availability, Atm. Chem. Phys., 16, 4579-4591.

Guo, H., A. P. Sullivan, P. Campuzano-Jost, J. C. Schroder, F. D. Lopez-Hilfiker, J. E. Dibb, J. L. Jimenez, J. A. Thornton, S. S. Brown, A. Nenes, and R. J. Weber (2016), Fine particle pH and the partitioning of nitric acid during winter in the northeastern United States, J. Geophys. Res., in review.

Guo, H., L. Xu, A. Bougiatioti, K. M. Cerully, S. L. Capps, J. R. Hite, A. G. Carlton, S.-H. Lee, M. H. Bergin, N. L. Ng, A. Nenes, and R. J. Weber (2015), Predicting particle water and pH in the southeastern United States, Atmos. Chem. Phys., 15, 5211–5228.

Hennigan, C. J., J. Izumi, A. P. Sullivan, R. J. Weber, and A. Nenes (2015), A critical evaluation of proxy methods used to estimate the acidity of atmospheric particles, Atm. Chem. Phys., 15, 1-15.

Weber, R. J., H. Guo, A. G. Russell, and A. Nenes (2016), High aerosol acidity despite declining atmospheric sulfate concentrations over the past 15 years, Nature Geoscience, 9(10.1038/ngeo2665), 282-285.

References

Ansari, A. S., and Pandis, S. N.: Response of inorganic PM to precursor concentrations, Environmental Science & Technology, 32, 2706-2714, 10.1021/es971130j, 1998.

Edgerton, E. S., Hartsell, B. E., Saylor, R. D., Jansen, J. J., Hansen, D. A., and Hidy, G. M.: The southeastern aerosol research and characterization study: Part II. Filter-based measurements of fine and coarse particulate matter mass and composition, Journal of the Air & Waste Management Association, 55, 1527-1542, 2005.

Flanagan, J. B., Jayanty, R. K. M., Rickman, E. E., and Peterson, M. R.: PM2.5 speciation trends network: Evaluation of whole-system uncertainties using data from sites with collocated samplers, Journal of   the Air & Waste Management Association, 56, 492-499, 2006.

Kim, P. S., Jacob, D. J., Fisher, J. A., Travis, K., Yu, K., Zhu, L., Yantosca, R. M., Sulprizio, M. P., Jimenez, J. L., Campuzano-Jost, P., Froyd, K. D., Liao, J., Hair, J. W., Fenn, M. A., Butler, C. F., Wagner, N. L., Gordon, T. D., Welti, A., Wennberg, P. O., Crounse, J. D., St Clair, J. M., Teng, A. P., Millet, D. B., Schwarz, J. P., Markovic, M. Z., and Perring, A. E.: Sources, seasonality, and trends of southeast US aerosol: an integrated analysis of surface, aircraft, and satellite observations with the GEOS-Chem chemical transport model, Atmospheric Chemistry and Physics, 15, 10411-10433, 10.5194/acp-15-10411-2015, 2015.

Park, R. J., Jacob, D. J., Field, B. D., Yantosca, R. M., and Chin, M.: Natural and transboundary pollution influences on sulfate-nitrate-ammonium aerosols in the United States: Implications for policy, Journal of Geophysical Research-Atmospheres, 109, 10.1029/2003jd004473, 2004.
* * *
**Response to Reviewer #2**

The authors present data suggesting that the sulfate in the Eastern US is not fully neutralized by ammonia despite the availability of the latter in the gas phase and argue that this could be due to mass transfer delays between the gas and particulate phase caused by organics. This is clearly an important issue that has received a lot of attention in the literature during the last few years. The paper is well written, however, the evidence provided to support its major conclusion is rather weak and it neglects other simpler explanations. It also neglects important work that has been published recently explaining similar observations with existing aerosol thermodynamics models. These problems are discussed below.

(1) Evidence that the aerosol is in equilibrium in the Southeast US The authors conclude from their data analysis that the aerosol in the eastern US is not in equilibrium probably due to limitations of the mass transfer of ammonia due to the organics. However, they neglect at least two studies that have used high quality gas-phase ammonia measurements in the same area and season to investigate this issue and reached the opposite conclusion. Weber et al. (2016) showed that the measured gas-phase ammonia was in equilibrium with the particulate phase during SOAS (see for example Figure 1 in that paper). They used 1-hr as averaging time therefore the equilibration timescale should be much less than this. If there was a significant delay in the mass transfer of ammonia between the two phases there should be significant observed discrepancies. Similar conclusions were reached by Nowak et al. (2006) for the 2002 ANARChE study in Atlanta. They concluded that the agreement between the measured and predicted ammonia within the uncertainties suggests that the assumption of thermodynamic equilibrium on the 7.5 min timescale is appropriate for most of the ANARChE data examined here.

We have amended the text to discuss the findings of Weber et al. (2016) (page 2 line 24-page 3 line 1) and identify the remaining need to reconcile simultaneous observations of low ammonium-sulfate ratios ($R$) and significant gas-phase ammonia. We have added a discussion of gas-phase ammonia predicted by thermodynamic models (page 4 lines 4-6; Figure 1, page 18) as well as model comparisons to ammonia observations (Figure 7, page 24). We address the findings of Nowak et al. (2006) on page 4 lines 9-11.

(2) Alternative explanations Weber et al. (2016) have argued that the incomplete neutralization of sulfate in the southeast US and the corresponding changes during the last 15 years are in general consistent with our current understanding of inorganic aerosol thermodynamics. They did not need to invoke delays in mass transfer and lack of equilibrium between the two phases. The authors appear to try something similar in Figure 3 but in a rather convoluted way using wet deposition data (please see comment 6 below).

See response to comment 1 above. Weber et al. (2016) cannot explain the low observed values of $R$ and we have now clarified this in the text. See our responses to reviewer 1.

(3) Model evaluation The use of GEOS-Chem to test the hypothesis of delays in the mass transfer of ammonia is the most original aspect of the current work. However, the evaluation of the corresponding GEOS-Chem predictions is rather superficial given the availability of the SOAS measurements during the same summer. The ability of GEOS-Chem to reproduce observed gas-phase ammonia concentrations is critical for the authors' argument. For example, the problems of the base case simulation could be due to the overprediction of ammonia availability (e.g., due to errors in ammonia emissions) and the assumed delay in mass transfer could be correcting one error by introducing another.

In response to the reviewer, we have added an evaluation of gas-phase ammonia in GEOS-Chem in the standard model and model with limited $NH_3$ uptake compared to observations from the SEARCH network (page 9 lines 18-22, 29-32; Figure 7, page 24). We point out that different ammonia emission inventories for the US agree to within 20% in summer (page 4 lines 27-28; Paulot et al., 2014). We now emphasize the thermodynamic inconsistency between low $R$ and high $NH_3(g)$ as a key original component of this work.

(4) Terminology and details about calculated quantities I found the terminology used in the paper quite confusing. For example, the terms "sulfate neutralization" and "aerosol neutralization" are used throughout the paper instead of the ammonium to sulfate molar or equivalent ratio. Sometimes the nitrate is also used sometimes it is not. In all cases the other cations that could, in principle at least, be neutralizing sulfate are not included in this neutralization ratio.

At the reviewer's suggestion we have eliminated the "neutralization" terminology throughout the text and have replaced it with "ammonium-sulfate ratios" defined as $R = [NH_4^+]/[S(VI)]$. When nitrate is included in this ratio, it is defined as $R_N = ([NH_4^+]-[NO_3^-])/[S(VI)]$ and stated explicitly. Discussion of the influence of other cations has been amended in the text (page 5 lines 22-26).

It is not clear if the different ratios shown throughout the paper correspond to PM1, PM2.5, PM10 or something else. This detail is critical given the importance of calcium, magnesium, etc., for the coarse particles.

The particle size of the measurements shown is now given on page 19 line 8 and the focus throughout the paper on fine particulate matter is now stated on page 2 lines 5-6.

It is also not clear how the authors estimate the averages of the different ratios. Do they average the concentrations and then estimate the ratios or do they estimate the ratios with some averaging time (daily?) and then average them?

Mean ratios are presented as the ratio of mean quantities, now stated in the text (page 4 line 25) and in the caption of Figure 2 (page 19 lines 12-13).

(5) Originality of hypothesis The same hypothesis regarding the organic aerosol role in mass transfer of ammonia in the southeast US has been presented Kim et al. (ACP, 2015) with a number of common authors in the two studies. This should be discussed in the introduction of the paper.

The text has been amended on page 3 lines 8-9 to address this point.

(6) Use of wet deposition data The authors use wet deposition measurements as practically a surrogate of total (gas and particulate) ammonia. This is rather tricky given that rainfall takes place during specific meteolorogical conditions, clouds can produce sulfate, and there are of course different wet removal efficiencies for ammonia and sulfate. Despite these problems, the paper does not address the potential biases that could be introduced in the analysis because of the use of these data. This could be one of the reasons for the differences between the conclusions here and those of other studies. I realize that gas-phase ammonia measurements exist only during specific field campaigns, but there are enough of them available both in the US and Europe. Use of these measurements is clearly preferable to the wet deposition data.

We have clarified our use of wet deposition data as it relates to emissions of ammonia and $SO_2$ on page 5 lines 3-6. We have also added a model comparison to gas-phase ammonia measurements (Figure 7, page 24).

(7) Role of organosulfates and crustal elements Some additional information is needed regarding the potential role of organosulfates and crustal elements for the present analysis. The authors appear to assume that they are negligible for the purposes of this work. However, some quantitative arguments are needed taking for example advantage of the SOAS measurements.

As stated in the text (page 3 lines 1-5, page 5 lines 22-26) organosulfates and crustal elements are present in low concentrations in the Southeast US summer based on observations. Organosulfates were also present in low concentrations during SOAS (Budisulistiorini et al., 2015; Hettiyadura et al., 2015; Rattanavaraha et al., 2016) and these references have been added in the text (page 3 lines 4-5). We have added a calculation of the effect of crustal components specifically for the data set that we analyze (page 5, lines 25-26).

(8) Some additional minor points:

Page 2, lines 2-4. This is clearly not true when there are other cations present.

Other cations are not present in high concentrations under the conditions examined in this paper as discussed in the response to the comment above.

Figure 2a is rather misleading given that most of the sulfur dioxide is emitted by point sources. It should probably be replaced with a table with the emissions in different regions of the US (e.g., southeast, northeast, etc.).

Emissions in the eastern US are stated in the text (page 4 line 30) as well as ratios for the Southeast and Northeast (page 6 line 32).

References

Budisulistiorini, S. H., Li, X., Bairai, S. T., Renfro, J., Liu, Y., Liu, Y. J., McKinney, K. A., Martin, S. T., McNeill, V. F., Pye, H. O. T., Nenes, A., Neff, M. E., Stone, E. A., Mueller, S., Knote, C., Shaw, S. L., Zhang, Z., Gold, A., and Surratt, J. D.: Examining the effects of anthropogenic emissions on isoprene-derived secondary organic aerosol formation during the 2013 Southern Oxidant and Aerosol Study (SOAS) at the Look Rock, Tennessee ground site, Atmospheric Chemistry and Physics, 15, 8871-8888, 10.5194/acp-15-8871-2015, 2015.

Hettiyadura, A. P. S., Stone, E. A., Kundu, S., Baker, Z., Geddes, E., Richards, K., and Humphry, T.: Determination of atmospheric organosulfates using HILIC chromatography with MS detection, Atmospheric Measurement Techniques, 8, 2347-2358, 10.5194/amt-8-2347-2015, 2015.

Paulot, F., Jacob, D. J., Pinder, R. W., Bash, J. O., Travis, K., and Henze, D. K.: Ammonia emissions in the United States, European Union, and China derived by high-resolution inversion of ammonium wet deposition data: Interpretation with a new agricultural emissions inventory (MASAGE_NH3), Journal of Geophysical Research-Atmospheres, 119, 4343-4364, 10.1002/2013jd021130, 2014.

Rattanavaraha, W., Chu, K., Budisulistiorini, H., Riva, M., Lin, Y. H., Edgerton, E. S., Baumann, K., Shaw, S. L., Guo, H. Y., King, L., Weber, R. J., Neff, M. E., Stone, E. A., Offenberg, J. H., Zhang, Z. F., Gold, A., and Surratt, J. D.: Assessing the impact of anthropogenic pollution on isoprene-derived secondary organic aerosol formation in PM2.5 collected from the Birmingham, Alabama, ground site during the 2013 Southern Oxidant and Aerosol Study, Atmospheric Chemistry and Physics, 16, 4897-4914, 10.5194/acp-16-4897-2016, 2016.

---

## Author Response (AR2)

We thank the anonymous reviewer for their helpful comments. The reviewer comments are in black and our responses are in blue and include page and line numbers where changes were made to the accompanying manuscript.

This paper invokes the idea of an organic film over an aqueous aerosol to explain observed low ammonium to sulfate molar ratios, despite excess gas phase ammonia. The paper has substantially changed since the first version, but again these authors chose to analyze aerosol pH effects through the use of molar ratios, which has been shown to be problematic.

The paper now focuses on the observation that both currently used thermodynamic models, ISORROPIA and E-AIM tend to predict higher molar ratios than what is observed. They have added some interesting plots (e.g., Fig 1 is quite nice) and arguments to support the idea that the lower molar ratios may be due to the organic film. Some issues should be considered and clarified in the text prior to publication.

A major issue is clarifying the logic of this hypothesis and its implications. If NH3 is not in equilibrium with the particle phase, the premise of this work, and given it is the most important fine mode base (in this case, ANS, the only base) it follows that the thermodynamic models cannot be used, as they assume equilibrium. The model predictions are then incorrect, including pH etc. Of coarse it raises the issue how the models do so well predicting actual NH3/NH4+ partitioning over a wide range of ambient conditions, something that should be explained (more on this below). Because this has very large implications, impacting many already published paper, this implication should be very clearly stated in the paper.

A second major point is the authors should explicitly state how the organic film impedes the uptake of NH3, is it through a low accommodation coefficient, or maybe low solubility of NH3 in the organic film? A curious thing is that H2O and NH3 have nearly identical molar masses making many of their properties similar (diffusivities, etc). How these two molecules could behave so differently when interacting with the film should be directly addressed. More details below.

More Details:

1). A simpler explanation for the model vs measured molar ratio discrepancy could simply be that modeled pH vs observed molar ratios respond differently to treaing the fine mode as a bulk property. When averaged over all sizes, observed molar ratios may be sensitive to this assumption, whereas actual pH less so. It has already been shown that molar ratios and pH are not necessarily related in a simple way. This possible explanation for the molar ratio discrepancy (or this complication when interpreting the data) should be noted in the paper.

We have added on page 3 lines 6-8: "The chemical composition of individual sulfate particles may deviate from the bulk, but it is not clear how such inhomogeneity could explain the observed departure from simple thermodynamics."

2) Issues with imprecise statements. Eg, in the Abstract, lines 18 to 22 (and other related discussions in the paper, eg first sections of Intro.), regarding ammonia concentrations and molar ratios. The statements are based on the extreme end-members of processes that are asymptotic (E-AIM solution in Fig 1), and the ambient condition is between the two extremes.

We add references to Zhang et al. (2002), Martin et al. (2004), and Yu et al. (2005) to support the statements that the thermodynamic behavior described is observed in a range of environments in the US (page 2 lines 13-14).

Line 29-30 statement that molar ratios decrease with decrease sulfate is incompatible with theory is, I believe, not correct, it is predicted by the model (see more on this below).

We amend the text to refer to Marais et al. (2016) that shows the significant decreasing trend in aerosol acidity predicted by ISORROPIA in response to decreasing sulfate, supporting the statement in the text (page 7 lines 25-26).

3) The authors assert the film will lower the rate of NH3 uptake so that equilibrium is not achieved since the time scales to reach equilibrium become long. This would seem to then invalidate the use of the thermodynamic models in general since the model assumes equilibrium for all semi-volatile species, including water vapor. To argue that an organic film affects NH3 mass transport dynamics, affects on other species should also be considered. The thermodynamics is a multi-component system, the various species behavior being inter-connected. For example, if NH3 time scales to reach equilibrium become large, what happens to other key species, such as H2O, HNO3, etc interacting with the organic surface. See discussion below on diffusivities of various species through an organic film affecting reactive uptake. The conclusion from below is that water will also have a similar time scale to reach equilibrium as ammonia, by the mechanism proposed here, and nitric acid will take about 2 times as long to reach equilibrium. How this will be handled in the CTM and how this is reconciled with existing data and publications on water uptake of particles, etc, should be specifically discussed, along with a detailed analysis of exactly how an organic film will affect reactive uptake. That is, exactly what is the mechanism that slows NH3, but not other species? Maybe it is differences in accommodation coefficients. If so, is this reasonable given measured accommodation coefficients? It should be explicitly stated what the possible physical explanation is for this proposed resistance to NH3 uptake.

See response to discussion of NH$_3$ uptake below.

4) pH matters, not molar ratios, when concerned about acidity effects. The goal of all this work, as noted by the authors in the abstract, is to predict the effects of aerosol pH on other aerosol properties or processes. Eg, from the abstract, where it states that … uptake of ammonia has important implications for aerosol mass, hygroscopicity, and acidity. It also states that … decrease sulfate aerosol will not have the co-benefit of suppressing acid-catalyzed secondary organic aerosol (SOA) formation. The key is aerosol pH, not molar ratios. As already noted, a reported analyses [Guo et al., 2016; Guo et al., 2015], suggests that in terms of partitioning of semi-volatile species, ISORROPIA works fairly well. That is, the models provide accurate

predictions of properties of interest related to pH, something that molar ratios cannot do, as far as I can tell.

We now clarify in the text (page 4 lines 23-25) that we do not use molar ratios as a proxy for pH and instead in this work we explore only the measureable quantities aerosol ammonium and sulfate.

5) One aspect the film achieves, according to this work, is that in the CTM model NH3 is predicted better (in addition to molar ratios, which is discussed above). Given the large uncertainties in NH3 emissions, could this simply be fortuitous? How robust a test of the model is the use of measured versus predicted NH3? Accurate prediction of gas particle partitioning of semi-volatile species would seem a better approach. For example, can the CTM accurately predict NH4+/NH3, HNO3/NO3- partitioning, as has been done with ISORROPIA (see various references)?

Paulot et al. (2014) showed there is good confidence in US ammonia emissions, which agree within 20% for independent bottom-up and top down estimates (page 4 lines 31-32). We have also added a comparison of modeled and observed ammonium wet deposition (page 9 lines 22-25) in order to show that uncertainties in emissions are not enough to explain the large model underestimate in gas-phase ammonia, so instead the problem is due to thermodynamic partitioning. As stated in Guo et al. (2016), the even distribution of ammonia between the gas and aerosol phase in summertime makes it the instructive quantity to test model partitioning while most nitrate is present in the gas phase in the Southeast US summer.

6) An explanation for the issue of decreasing trends of sulfate and molar ratios discussed in this manuscript is given elsewhere [Weber et al., 2016]. It can simply be explained by ammonia volatility, which should be pointed out in this paper.

We state in the text (page 7 lines 18-20, 27-29) the consistency of observed trends in sulfate and molar ratios shown in Weber et al. (2016) and we have added the explanation presented by Weber et al. (2016) (page 2 line 32).

Mass Transport Limitations in Aqueous-Phase Chemistry (the following is based on Seinfeld and Pandis, 2nd Edition, 2006, Chapter 12, Section 12.2).

In trying to understand the physical details associated with the proposed interaction between NH3 and an organic film, the following analysis of characteristic times for equilibrium was undertaken, following the argument put forward in the paper. A comparison is made between NH3 and H2O since they have nearly identical molecular weights.

If the rate-limiting step for mass transfer of a gas, say NH3 in this case, from the gas to the particle surface and then through the organic film to the bulk water of the particle interior, (ultimately leading to conversion of NH3 to NH4+), is gas transport through an organic film on the perimeter of the particle, the characteristic time ($\tau$) for the system to reach equilibrium will be in proportion to:

τNH3 α Rp2/(γ DNH3-aerosol), (from S & P equation 12.61)

where Rp could be thought of as the thickness of the film, γ the accommodation coefficient and DNH3-aerosol the diffusivity of NH3 in the organic layer. If we assume that the gases of interest will all have fairly similar accommodation coefficients, all being fairly sticky, and effective Henry's law constants (similar solubility in the OA phase) the time scales comes down to diffusivity (S&P Eq 12.62), or roughly the ratio depends on sqrt(molecular weight) of the solute (Eq 12.61).

Water (18 g/mole) has a very similar molecular weight as NH3 (17 g/mole) so time scales to reach equilibrium for these two species should be roughly equal. The ratio of equilibration time scales of NH3 and H2O due to the organic film is very roughly then = sqrt(18/17) = 1.03, or approximately 1. Thus, with similar accommodation coefficients and solubility in the OA, if NH3 is not in equilibrium due to the film, water is also not, and will behave very similar to NH3. This would contradict many studies demonstrating that in the ambient atmosphere water is in equilibrium, thus NH3 should also be, by this analyses, or the assumptions are wrong.

Consider a different semi-volatile species, say HNO3 (molecular weight=63 g/mole). In this case the time scales for equilibrium for HNO3 relative to NH3 will be sqrt(63/17) = 1.9. Thus nitric acid takes about 2 times as long to reach equilibrium. How can this be reconciled based on ambient data showing good agreement with measured partitioning of nitric acid and nitrate under most conditions (eg, [Guo et al., 2016])? How would this be included in the overall CTM?

Note that all other semi-volatile species of importance when assessing aerosol pH (e.g, HCl MW=36 g/mole, organic acids…) will have larger molecular weights than NH3, so it will take even longer for them to reach equilibrium. Maybe this all can be explained by differences in mass accommodation coefficients? In fact, it seems the only plausible way.

If possible, the authors should attempt to explicitly state in the text by what mechanism the NH3 is selectively impeded, as these could be measured to test the hypothesis of the paper. Overall, the situation is getting very complicated relative to a straight thermodynamic analysis (assumes equilibrium) that appears to agree with observations of liquid water content and partitioning of semi-volatile species sensitive to pH.

Liggio et al. (2011) shows that the predicted characteristic timescale (for uptake controlled by accommodation and diffusion) for equilibrium using an accommodation coefficient of 1.0 is only appropriate for the organic-free experiments while the measured timescale for equilibrium for experiments with ambient particles are on the order of hours rather than seconds. We note this point and as we now cite Daumer et al. (1992) that also showed a retardation of the ammonium-sulfate equilibrium when sulfuric acid particles are coated with organic films (page 8 lines 5-8).

Laboratory measurements of organic aerosol (Wong et al., 2014) and observations of ambient particles (Raatikainen et al., 2013) have shown the accommodation coefficient for water is > 0.1 meaning it is above the threshold where a kinetic limitation is expected to have an effect on particle growth. Additionally, while we do not explore the effect in this work, Liu et al. (2015) has shown that reactions between ammonia and organics can occur and even compete with the

uptake of ammonia by acidic sulfate aerosol. As the most abundant source of gas-phase reduced nitrogen, we would expect the formation of nitrogen-containing organic fragments to be important for ammonia but not for nitrate. We have added these points to the Conclusions (page 10 line 32-page 11 line 1, page 11 lines 4-6).

We also now discuss in the Conclusions the work of Guo et al. (2016) on the partitioning of wintertime nitrate and underscore the need for more work to assess the impact of organics on the partitioning of water and semi-volatile species (page 11 lines 3-8).

Guo, H., A. P. Sullivan, P. Campuzano-Jost, J. C. Schroder, F. D. Lopez-Hilfiker, J. E. Dibb, J. L. Jimenez, J. A. Thornton, S. S. Brown, A. Nenes, and R. J. Weber (2016), Fine particle pH and the partitioning of nitric acid during winter in the northeastern United States, J. Geophys. Res. Atmos., 121(17), 10,355-310,376.

Guo, H., L. Xu, A. Bougiatioti, K. M. Cerully, S. L. Capps, J. R. Hite, A. G. Carlton, S.-H. Lee, M. H. Bergin, N. L. Ng, A. Nenes, and R. J. Weber (2015), Predicting particle water and pH in the southeastern United States, Atmos. Chem. Phys., 15, 5211–5228.

Weber, R. J., H. Guo, A. G. Russell, and A. Nenes (2016), High aerosol acidity despite declining atmospheric sulfate concentrations over the past 15 years, Nature Geoscience, 9(10.1038/ngeo2665), 282-285.

References

[revised manuscript text omitted]